# CRRC: Residual Cross-view Learning for Deep Multi-view Clustering

## Abstract

Deep multi-view clustering aims to integrate complementary information from multiple heterogeneous views to improve clustering performance. However, existing fusion strategies often struggle to balance shared semantics and view-specific heterogeneity, as they typically rely on direct concatenation or rigid alignment, which obscures subtle cross-view patterns and assumes equal contribution from all views. To address these limitations, we propose CRRC, a novel framework that leverages residual connections to recalibrate view-specific features by adaptively incorporating complementary information from other views. Specifically, CRRC introduces a dynamic gating fusion module to control residual flow based on view characteristics, and an attention-based weighting mechanism to emphasize semantically relevant cross-view signals. These components work collaboratively to enhance feature discriminability and consistency. Extensive experiments on benchmarks datasets demonstrate that CRRC outperforms state-of-the-art methods in accuracy, NMI, and purity, validating its effectiveness in achieving robust multi-view clustering.

## 1 Introduction

Deep Multi-View Clustering (DMVC) employs deep neural networks to extract rich view-specific feature representations and model complementary dependencies across heterogeneous modalities like images, texts, and sensor data. Unlike simple pfeature concatenation, DMVC captures non-linear relationships and semantic diversity more effectively. Recent advances leveragep adversarial alignment (Huang et al., 2024), generatpive learning (Palumbo et al., 2024), and contrastive objectives (Sun et al., 2024) to improve robustness against noise and view imbalance, excelling in tasks such as cross-modal retrieval and multi-source perception.

DMVC methods mainly fall into two-stage (Du et al., 2021; Wang et al., 2024; Zhao et al., 2017; Peng et al., 2019) and end-to-end frameworks (Xu et al., 2024; Xia et al., 2022; Jiang et al., 2024; Zheng et al., 2024). Two-stage methods optimize feature learning and clustering separately, risking task misalignment, while end-to-end models jointly optimize both, including contrastive-, generative-, and clustering-integrated approaches. Designing semantic spaces is challenging: some enforce global alignment in a shared latent space (Tian et al., 2020; Hu et al., 2023), others preserve view-specific features via independent encoders (Xu et al., 2023). Balancing semantic consistency and modality diversity remains critical (Yan et al., 2023; Chen et al., 2025).

This study addresses two key challenges in current multi-view fusion methods. First, existing fusion strategies often struggle to balance shared semantics and view-specific heterogeneity. Many methods perform fusion by directly concatenating or aligning multi-view representations, which tends to obscure complementary patterns and weaken the uniqueness of each view. Despite efforts using adversarial learning (Yang et al., 2024) or decoupled autoencoders (Xu et al., 2021) to alleviate these issues, they often treat all views uniformly using fixed rules or shared transformations, overlooking the diverse semantic roles that different views may play, and lack a fine-grained and adaptive mechanism to selectively integrate complementary information. Second, by adopting uniform and static weighting strategies, some methods implicitly assume equal contribution from all views, failing to account for view-specific differences in informativeness and irrelevant information, thereby undermining the effectiveness of feature fusion. Although some approaches introduce learned weighting schemes or optimization-based strategies to mitigate this issue (Chen et al., 2025; Khalafaoui et al.,

2026; Jiang et al., 2026), they often lack the ability to filter out redundant or irrelevant content, neglect fine-grained semantic alignment across views, and fail to effectively model inter-view dependencies, resulting in suboptimal representations.

In this paper, we propose CRRC, a novel multi-view fusion framework designed to learn a discriminative shared semantic space across multiple views through contrastive learning. To achieve this, we perform independent mapping for each view to preserve their inherent heterogeneity. Instead of directly fusing features, CRRC selectively introduces residual components that represent cross-view differences. This strategy maintains the specificity of each view while strategically enabling cross-view interaction, and the residual connections ensure stable gradient flow during training. To further control this fusion, we design a dynamic gating fusion module, where a learnable gating vector adaptively modulates the residual flow, ensuring a flexible trade-off between shared semantics and private features. Additionally, we introduce an attention-based cross-view weighting module, which computes attention scores from other views to the current view, helping the model focus on more informative cross-view signals while suppressing irrelevant and redundant information. All components collaboratively construct a comprehensive and discriminative representation that serves as the anchor for contrastive learning. This enables the model to align semantically similar instances across views while distinguishing dissimilar ones, thereby enhancing clustering performance. Importantly, CRRC proposes a novel paradigm of residual recalibration combined with contrastive learning co-optimization, which synergistically integrates view-specific preservation and cross-view semantic alignment, rather than simply introducing residual components. Empirical results demonstrate that this joint design significantly outperforms using individual modules or alternative learning objectives alone.

In brief, the main contributions of this paper are summarized as follows:

- We propose a cross-view recalibration framework via residual connections, which combines the current view with aggregated complementary information in a residual manner to effectively model cross-view increments.

- We design a dynamic gating fusion module, which adaptively controls the strength of residual components based on the current view's characteristics, enabling flexible regulation of complementary information flow.

- We introduce an attention-based cross-view weighting module, which dynamically captures inter-view correlation differences to enhance the semantic consistency of the fused representation.

## 2 RELATED WORK

### 2.1 TWO-STAGE DEEP MULTI-VIEW CLUSTERING

Two-stage DMVC methods decouple feature learning and clustering, often leading to suboptimal performance due to misaligned objectives. Autoencoder-based approaches (Xu et al., 2021; Du et al., 2021) extract view-specific embeddings using AEs or VAEs, followed by clustering algorithms like $k$-means. Subspace methods (Wang et al., 2024; Zhu et al., 2024) impose low-rank or sparse constraints to learn shared structures. Matrix factorization-based models (Zhao et al., 2017; 2020; Zhang et al., 2018; Xing et al., 2019) decompose multi-view data into interpretable components for consensus clustering. Graph-based approaches (Nie et al., 2017; Peng et al., 2019) employ Graph Neural Networks (GNNs) or graph autoencoders to capture inter-view relationships via topological structures.

### 2.2 END-TO-END LEARNING PARADIGMS

End-to-end DMVC methods jointly optimize representation learning and clustering, enhancing cross-view coherence. Embedding-based approaches (Xu et al., 2024; Xia et al., 2022) use reconstruction and clustering losses (e.g., KL divergence, pseudo-labeling) to learn compact latent spaces, but require careful design to prevent collapse. Graph-based models (Du et al., 2023; Peng et al., 2019) integrate node attributes and structures via GNNs or GAEs, with hierarchical fusion or adaptive edge weights. Adversarial methods (Huang et al., 2024; Yang et al., 2024) align view

distributions using discriminators and auxiliary constraints (e.g., cycle-consistency, Wasserstein distance), improving robustness but potentially over-aligning view-specific semantics.

### 2.3 CONTRASTIVE MULTI-VIEW CLUSTERING

Contrastive learning improves discriminative feature learning by pulling positives close and pushing negatives apart. Vanilla methods (Hassani & Khasahmadi, 2020; Chen et al., 2020) align different views of the same instance to enhance semantic consistency. To mitigate false negatives, robust variants (Cui et al., 2024; Wang & Feng, 2024; Guo et al., 2024) use global affinity or probabilistic correction. Multi-level contrast methods (Zhang & Che, 2024; Wang et al., 2023; Bian et al., 2025; Chen et al., 2025) integrate instance-, feature-, and cluster-level signals to avoid collapse. Structure-guided contrast methods (Fei et al., 2025; Guo et al., 2026; Cui et al., 2026) introduce local structural information to enhance the reliability of positive sample pairs. However, most contrastive models lack fine-grained control for adaptively fusing complementary cross-view information.

## 3 METHOD

### 3.1 OVERALL WORKFLOW

As illustrated in Figure 1, our proposed CRRC framework follows an end-to-end workflow, which consists of the following key steps: (1) Multi-view Encoding and Reconstruction: The input data $\mathbf{X}^v$ for each view is encoded by its dedicated autoencoder to obtain the view-specific latent representations $\mathbf{Z}^v$, while simultaneously being reconstructed to ensure feature quality. (2) Global and Local Representation Generation: All $\mathbf{Z}^v$ are concatenated and fused by a shared Multilayer Perceptron (MLP) to produce the unified global semantic representation $\mathbf{H}$. Concurrently, each $\mathbf{Z}^v$ is processed by a view-specific MLP to generate the local representation $\mathbf{R}^v$ for fine-grained recalibration. (3) Residual Recalibration: For each view $v$, the recalibration process consists of three steps:

- Attention-based Weighting: Using the global representation $\mathbf{H}$ as the query, attention weights $\alpha^{k \neq v}$ are computed over all other views' representations $\mathbf{R}^{(k \neq v)}$, which are aggregated into a weighted complementary information vector.
- Dynamic Gating: A dynamic gating vector $\mathbf{g}_v$ is generated based on the current view $\mathbf{R}^v$ and the cross-view complementary information.
- Residual Fusion: The weighted complementary information is modulated by the gating vector $\mathbf{g}_v$ to form the residual term, which is added to the original feature to obtain the recalibrated representation $\hat{\mathbf{R}}^v$.

(4) Contrastive Learning and Clustering: The global representation $\mathbf{H}$ and all recalibrated features $\{\hat{\mathbf{R}}^v\}$ are used for cross-view contrastive learning, pulling together different views of the same instance while pushing apart those from different instances. Finally, $\mathbf{H}$ is used for $k$-means clustering to produce the final cluster assignments $\mathbf{Y}$.

### 3.2 MULTI-VIEW DATA RECONSTRUCTION

In multi-view clustering, data from different views often contain heterogeneity and view-specific noise. To address this, CRRC employs independent autoencoders for each view to extract low-dimensional, discriminative features while preserving essential information. Given a dataset with $m$ views and $n$ samples, denoted as $\{\mathbf{X}^v = \{\mathbf{x}_1^v, \ldots, \mathbf{x}_n^v\} \in \mathbb{R}^{n \times D_v}\}_{v=1}^m$ where $\mathbf{x}_i^v \in \mathbb{R}^{D_v}$ is the $i$-th sample from view $v$. Each view is encoded using a dedicated encoder $f_\theta^{(v)}(\cdot)$ and reconstructed by a corresponding decoder $g_\phi^{(v)}(\cdot)$:

$$\mathbf{z}_i^v = f_\theta^v(\mathbf{x}_i^v), \quad \hat{\mathbf{x}}_i^v = g_\phi^v(\mathbf{z}_i^v), \tag{1}$$

where $\mathbf{z}_i^v \in \mathbb{R}^{d_v}$ is the latent representation and $\hat{\mathbf{x}}_i^v$ denotes its reconstruction. To ensure semantic preservation and reduce redundancy, a reconstruction loss is applied:

$$\mathcal{L}_{\text{rec}} = \sum_{v=1}^m \sum_{i=1}^n \|\mathbf{x}_i^v - \hat{\mathbf{x}}_i^v\|_F^2, \tag{2}$$

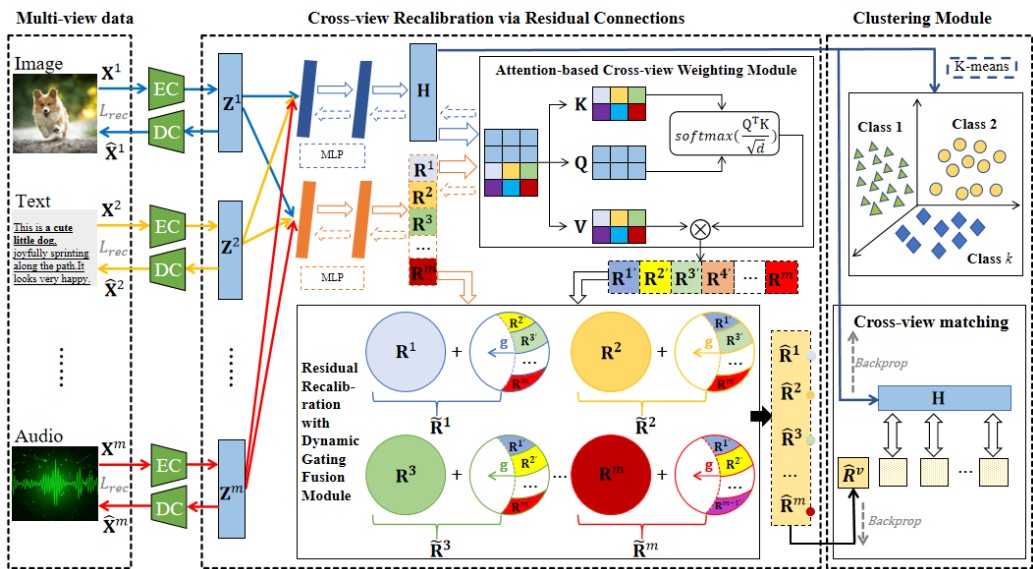

Figure 1: The CRRC framework encodes multi-view inputs to retain shared and view-specific information. A global representation is aggregated from latent features, while residual connections recalibrate view-specific representations under dynamic gating and attention-based weighting. Contrastive learning aligns them for effective clustering.

which encourages the encoder to learn compact, representative features, providing a solid basis for subsequent fusion and contrastive learning.

## 3.3 RESIDUAL RECALIBRATION WITH DYNAMIC GATING FUSION MODULE

This section introduces a residual recalibration strategy guided by a dynamic gating mechanism. Instead of directly fusing features from multiple views, we model their differences via residual components, which serve to capture complementary cross-view information. A learnable gating vector is then used to adaptively control the flow of these residuals based on the view-specific characteristics. This design enables flexible and selective feature integration, while maintaining stable optimization through residual connections.

Given the embedded representations from $m$ views as $\{\mathbf{Z}^v = \{\mathbf{z}_1^v, \mathbf{z}_2^v, \ldots, \mathbf{z}_n^v\}\}_{v=1}^m$, where $\mathbf{z}_i^v \in \mathbb{R}^{d_v}$ denotes the representation of the $i$-th sample in the $v$-th view, we first concatenate all view-specific features and feed them into a shared MLP to generate a unified global semantic representation:

$$\mathbf{H} = \mathcal{F}_{\mathrm{MLP}}\left(\left[\mathbf{Z}^1 \oplus \mathbf{Z}^2 \oplus \cdots \oplus \mathbf{Z}^m\right]; \Phi\right), \tag{3}$$

where $\oplus$ denotes feature concatenation and $\Phi$ represents the parameters of the global fusion MLP. We obtain $\mathbf{H} \in \mathbb{R}^{n \times d}$, where $n$ is the number of samples and $d$ is the feature dimension. In parallel, to capture complementary semantics specific to each view, each $\mathbf{Z}^v$ is further processed by a view-specific MLP with parameters $\Psi^v$ to produce the view-specific representation:

$$\mathbf{R}^v = \mathcal{F}_{\mathrm{MLP}}(\mathbf{Z}^v; \Psi^v) \in \mathbb{R}^{n \times d}. \tag{4}$$

Additionally, to enhance the representation capability of each view-specific feature representation and effectively integrate complementary information across views, we introduce a residual recalibration mechanism for each view-specific feature representation $\mathbf{R}^v$, consisting of two main components: identity preservation and complementary integration. The identity path retains $\mathbf{R}^v$ directly in the final output, ensuring that view-specific information remains intact. In parallel, the integration path modulates $\mathbf{R}^v$ using residual information from the remaining views $\mathbf{R}^{(k \neq v)}$ to recalibrate its semantic representation:

$$\tilde{\mathbf{R}}^v = \mathbf{R}^v + \mathcal{M}(\mathbf{R}^v, \mathbf{R}^{(k \neq v)}), \tag{5}$$

where $\mathcal{M}(\cdot)$ denotes a residual modulation function that adaptively integrates cross-view information. Specifically, we implement this function as:

$$\mathcal{M}(\mathbf{R}^v, \mathbf{R}^{(k \neq v)}) = g_v \odot \mathcal{G}(\mathbf{R}^{(k \neq v)}), \tag{6}$$

where $\mathcal{G}(\cdot)$ is a cross-view semantic aggregator that dynamically gathers relevant information from the remaining views to support semantic alignment, which will be detailed in section 3.4.

To enable adaptive recalibration, it is essential to generate effective gating weights that dynamically control the extent of cross-view integration. These dynamic gates help disentangle view-specific and complementary feature representations, making the recalibration process more context-aware. Therefore, we define a learnable gating vector in Eq. (6) as:

$$g_v = \sigma\Big(\mathbf{W}_2\,\delta\big(\mathbf{W}_1[\mathbf{R}^v;\,\mathbf{R}^{(k \neq v)}] + b_1\big) + b_2\Big), \tag{7}$$

where $\delta(\cdot)$ denotes the ReLU activation function and $\sigma(\cdot)$ is the sigmoid function. $(\mathbf{W}_1 \in \mathbb{R}^{h \times 2d},\ b_1 \in \mathbb{R}^h)$, $(\mathbf{W}_2 \in \mathbb{R}^{d \times h},\ b_2 \in \mathbb{R}^d)$ are the weights and biases of the two linear transformations. The hidden dimension $h$ controls the gating capacity, capturing richer interactions between $\mathbf{R}^v$ and $\mathbf{R}^{(k \neq v)}$. This two-layer gating structure constitutes the core of the dynamic gating fusion module, which adaptively modulates the integration strength of cross-view residuals.

The dynamic gating vector $g_v \in [0,1]^{n \times d}$ adjusts based on input content, selectively controlling residual flow. Unlike static fusion methods, this dynamic gating mechanism can adaptively suppress the influence of irrelevant views, thereby enhancing the discriminability of representations. From an information-theoretic perspective, the gating operation reduces redundancy by filtering out less informative components, which helps strengthen semantic focus. Notably, the sigmoid-based gating allows gradients to backpropagate through both $\mathbf{R}^v$ and $\mathbf{R}^{(k \neq v)}$, thereby mitigating vanishing gradients, while the residual connection provides an additional shortcut that improves training stability. The related proofs are provided in Appendix A.1. This design ensures that even if residual contributions are strongly suppressed, the original view-specific information $\mathbf{R}^v$ is preserved through the identity path, effectively preventing complete information loss and providing a robust lower bound for feature quality. With these properties, the residual-fused recalibrated feature representation is then formulated as:

$$\tilde{\mathbf{R}}^v = \mathbf{R}^v + g_v \odot \mathcal{G}(\mathbf{R}^{(k \neq v)}). \tag{8}$$

## 3.4 ATTENTION-BASED CROSS-VIEW WEIGHTING MODULE

While the dynamic gating fusion module selectively regulates residual features, it does not explicitly account for the varying semantic importance of complementary views. To enhance this process, we introduce an attention-based cross-view weighting module that dynamically estimates cross-view semantic relevance and modulates residual contributions accordingly.

Let $\mathbf{R}^v$ denote the current view feature representation and $\mathbf{R}^{(k \neq v)}$ the complementary view feature representations. We implement the semantic aggregator $\mathcal{G}(\cdot)$ in Eq. (8) as a learnable attention-based fusion function. The complementary feature representations serve as keys and values, while the global semantic representation $\mathbf{H}$ is projected as the query $\mathbf{Q} = \mathbf{W}_q\mathbf{H}$ with learnable $\mathbf{W}_q$. The attention weights are computed via scaled dot-product between the query and each complementary view:

$$\alpha^{k \neq v} = \frac{\exp(\mathbf{R}^{(k \neq v)\top}\mathbf{Q})}{\sum_{\substack{k=1 \\ k \neq v}}^{m} \exp(\mathbf{R}^{(k \neq v)\top}\mathbf{Q})}, \tag{9}$$

where $\alpha^{(k \neq v)}$ reflects the importance of view $k$ in recalibrating current view $v$ and $\sum_{\substack{k=1 \\ k \neq v}}^{m} \alpha^{k \neq v} = 1$.

The fusion function $\mathcal{G}(\cdot)$ aggregates residuals weighted by attention:

$$\mathcal{G}(\mathbf{R}^{(k \neq v)}) = \sum_{\substack{k=1 \\ k \neq v}}^{m} \alpha^{(k \neq v)}\mathbf{R}^{(k \neq v)}. \tag{10}$$

This enables dynamic alignment, routing informative information from semantically related views and suppressing less relevant ones. The attention-based cross-view weighting module models inter-view correlations, providing fine-grained view selection to improve reliability and discriminability.

Finally, the recalibrated feature representation for the $v$-th view, originally defined in Eq. (8), is now evolved into the following form:

$$\hat{\mathbf{R}}^v = \mathbf{R}^v + g_v \odot \sum_{\substack{k=1 \\ k \neq v}}^{m} \alpha^{(k \neq v)} \mathbf{R}^{(k \neq v)}. \tag{11}$$

To further enhance the discriminative capability and cross-view alignment of the learned feature representations, we adopt a contrastive learning strategy based on the recalibrated feature representations. For each sample $i$, the global semantic representation $\mathbf{H}_i$ and the recalibrated feature $\hat{\mathbf{R}}_i^v$ from the same view form a positive pair, while mismatched pairs $(\mathbf{H}_i, \hat{\mathbf{R}}_j^v)$ with $j \neq i$ serve as negatives. We measure the similarity of each pair using cosine similarity, $\mathrm{Sim}(\mathbf{H}_i, \hat{\mathbf{R}}_j^v) = \frac{\langle \mathbf{H}_i, \hat{\mathbf{R}}_j^v \rangle}{\|\mathbf{H}_i\| \cdot \|\hat{\mathbf{R}}_j^v\|}$. The contrastive loss for view $v$ is formulated as:

$$\mathcal{L}_{\mathrm{cm}}^v = -\frac{1}{n} \sum_{i=1}^{n} \log \frac{\exp\left(\mathrm{Sim}(\mathbf{H}_i, \hat{\mathbf{R}}_i^v)/\tau\right)}{\sum_{j=1}^{n} \exp\left(\mathrm{Sim}(\mathbf{H}_i, \hat{\mathbf{R}}_j^v)/\tau\right)}, \tag{12}$$

where $\tau$ is a temperature hyperparameter controlling the sharpness of the similarity distribution. Minimizing this loss aligns recalibrated local feature representations with their corresponding global semantic representation while separating features from different samples, enhancing cross-view consistency and view-specific discriminability for more expressive multi-view embeddings.

### 3.5 TIME COMPLEXITY ANALYSIS

The time complexity of CRRC primarily arises from the following components, which depend on the number of views $m$ and the number of samples $n$. (1) Autoencoder Forward Pass: $\mathcal{O}(mnD_v d)$, where $D_v$ is the input dimension of view $v$ and $d$ is the latent dimension. (2) Global Representation $\mathbf{H}$ Generation: This step concatenates features from all views and forwards them through a global MLP, resulting in a computational complexity of $\mathcal{O}(n(md)d_h)$, where $d_h$ denotes the hidden dimension of the global MLP. (3) Attention Module: The attention computation measures similarity between the global representation $\mathbf{H}$ and the representations $\mathbf{R}^k$ from the remaining $(m-1)$ views, with complexity $\mathcal{O}(m^2 nd^2)$. Since $m$ is typically small (e.g., 2–5), this cost remains manageable. (4) Dynamic Gating Module: The dynamic gating vector for each view is produced using a two-layer MLP, leading to a complexity of $\mathcal{O}(nmdh_g)$, where $h_g$ is the hidden dimension of the gating network. (5) Contrastive Loss: Computing pairwise similarities across all samples in a batch incurs a full complexity of $\mathcal{O}(mn^2 d)$, which is the dominant cost. With mini-batch training, the practical complexity reduces to $\mathcal{O}(mB^2 d)$, where $B$ is the batch size (set to 256 in our experiments), which aligns with standard practice in contrastive learning.

### 3.6 OPTIMIZATION AND CLUSTERING

The overall training of our CRRC framework is guided by a joint objective that balances reconstruction fidelity with semantic consistency and cross-view feature alignment. Specifically, the total loss is defined as:

$$\mathcal{L}_{\mathrm{total}} = \mathcal{L}_{\mathrm{rec}} + \mathcal{L}_{\mathrm{cm}} = \mathcal{L}_{\mathrm{rec}}\left(\{\mathbf{X}^v, \hat{\mathbf{X}}^v\}_{v=1}^{m}; \{\theta^v, \phi^v\}_{v=1}^{m}\right) + \mathcal{L}_{\mathrm{cm}}\left(\{\hat{\mathbf{R}}^v, \mathbf{H}\}_{v=1}^{m}; \Psi, \Phi, \{\theta^v\}_{v=1}^{m}\right). \tag{13}$$

The overall training procedure is summarized in Algorithm 1.

By learning inherently cluster-friendly global features through joint reconstruction and contrastive objectives, we can directly apply simple clustering methods $k$-means on the global semantic representation $\mathbf{H}$ without relying on explicit clustering losses, thereby avoiding instability from inaccurate pseudo-labels. Formally, the final clustering assignment is obtained by optimizing the standard matrix-form $k$-means objective:

$$\mathbf{Y} = \arg\min_{\mathbf{Y}} \|\mathbf{H} - \mathbf{Y}\mathbf{C}\|_F^2, \tag{14}$$

where $\mathbf{Y} \in \{0,1\}^{n \times K}$ is a one-hot assignment matrix and $\mathbf{C} \in \mathbb{R}^{K \times d}$ contains the cluster centroids. Each row of $\mathbf{Y}$ assigns a data point to the closest cluster based on its refined global feature $\mathbf{H}_i$, and $\|\cdot\|_F$ denotes the Frobenius norm.

---

**Algorithm 1** Optimization of CRRC

---

**Input**: Multi-view data $\{\mathbf{X}^v\}_{v=1}^m$; Number of clusters $K$; Temperature coefficient $\tau$
**Output**: Cluster labels $\mathbf{Y}$

1: Pretrain autoencoders $f_\theta^{(v)}(\cdot)$ and decoders $g_\phi^{(v)}(\cdot)$ for each view by minimizing Eq. (2).
2: **while** not reaching max iterations $T_{\max}$ **do**
3:     Compute $\mathbf{Z}^v$ via Eq. (1).
4:     Compute $\mathbf{H}$ and $\{\mathbf{R}^v\}_{v=1}^m$ according to Eq. (3) and Eq. (4).
5:     **for** each view $v = 1$ to $m$ **do**
6:         Compute dynamic gating vectors for complementary views via Eq. (7).
7:         Compute attention-weighted residuals from complementary views using Eq. (10).
8:         Obtain recalibrated feature representations for the $v$-th view via Eq. (11).
9:     **end for**
10:    Compute contrastive loss as in Eq. (12).
11:    Compute total loss using Eq. (13).
12:    Update parameters via backpropagation using Adam optimizer.
13: **end while**
14: Derive final cluster labels from $\mathbf{H}$ using Eq. (14).
15: **return** $\mathbf{Y}$

---

# 4 EXPERIMENTS

## 4.1 EXPERIMENTAL SETTINGS

### 4.1.1 DATASETS DESCRIPTION AND COMPARISON METHODS

To evaluate our method, we conduct experiments on eight widely used multi-view datasets, covering both standard and large-scale benchmarks. NGs (Hussain et al., 2010) (500 documents, 3 views from different preprocessing methods, 5 classes); Fashion (Xiao et al., 2017) (three-view clothing images, 10,000 samples, 10 classes); MNIST-USPS (Peng et al., 2019) (two-view digits, 5,000 samples, 10 classes); Caltech-XV (Fei-Fei et al., 2004) (X=3,4,5 visual views) to test scalability across different view settings; Animal (Yang et al., 2022) (10,158 samples, 2 views, 50 classes) and Cifar100 (Krizhevsky et al., 2009) (50,000 samples, 3 views, 100 classes).

To comprehensively evaluate CRRC, we compare it with eleven recent DMVC methods spanning semantic enhancement and contrastive learning strategies. MFLVC (Xu et al., 2022) captures shared semantics via independent optimization across feature spaces. CVCL (Chen et al., 2023) contrasts cluster assignments to exploit label-level consistency. GCFAgg (Yan et al., 2023) reduces noise by leveraging sample complementarity across views. DCMVC (Cui et al., 2024) enhances discriminability through dual contrastive learning with cluster diffusion and neighbor alignment. HFMVC (Jiang et al., 2024) adapts to view heterogeneity in federated clustering. CSOT (Zhang et al., 2024) applies optimal transport to reweight semantic alignment. AccMVC (Yan et al., 2024) balances reconstruction and consistency via anchor-based contrastive learning. SCMVC (Wu et al., 2024) uses hierarchical fusion to emphasize informative views and suppress noise. DDMVC (Xu et al., 2025) promotes robust clustering by jointly modeling consistency and diversity. DMVC_MIC (Cui et al., 2025) employs meta-learning with information compression and semantic puzzles to produce compact, clustering-friendly representations. SparseMVC (Liu et al., 2025) mitigates cross-view sparsity and encoding inconsistencies through adaptive sparse autoencoders and correlation-based sample reweighting. These methods form a strong benchmark for assessing CRRC's effectiveness.

### 4.1.2 EVALUATION METRICS AND IMPLEMENTATION DETAILS

Clustering quality is evaluated by three commonly used metrics: clustering accuracy (ACC), normalized mutual information (NMI), and purity (PUR). These metrics quantify the consistency between the predicted cluster assignments and the true class labels, with larger values indicating superior clustering results.

Table 1: Clustering performance of all compared DMVC methods, where the best performance is highlighted in bold and the second best is underlined.

| Datasets | NGs | | | Fashion | | | MNIST-USPS | | | Caltech-3V | | |
|---|---|---|---|---|---|---|---|---|---|---|---|---|
| metrics | ACC | NMI | PUR | ACC | NMI | PUR | ACC | NMI | PUR | ACC | NMI | PUR |
| MFLVC (2022) | 86.80 | 74.27 | 86.80 | 99.20 | 98.00 | 99.20 | 99.60 | 98.84 | 99.60 | 60.57 | 57.61 | 61.14 |
| CVCL (2023) | 85.60 | 69.27 | 85.60 | 99.24 | 98.06 | 99.24 | 99.58 | 98.77 | 99.58 | 64.80 | 53.93 | 65.20 |
| GCFAgg (2023) | 81.20 | 75.85 | 81.20 | 89.28 | 93.14 | 89.28 | 99.54 | 98.71 | 99.54 | 63.57 | 53.92 | 63.57 |
| DCMVC (2023) | 62.00 | 56.46 | 62.00 | 96.35 | 93.96 | 96.35 | 89.20 | 86.46 | 89.20 | 61.71 | 44.55 | 65.00 |
| HFMVC (2024) | 83.93 | 64.73 | 83.93 | 92.68 | 89.76 | 92.68 | 98.10 | 96.00 | 98.00 | 68.21 | 58.80 | 68.79 |
| CSOT (2024) | 73.00 | 62.05 | 73.00 | 99.07 | 97.70 | 99.07 | 99.62 | 98.34 | 99.62 | _72.71_ | _62.25_ | _72.71_ |
| AccMVC (2024) | 91.20 | 78.39 | 91.20 | 99.16 | 97.84 | 99.16 | 99.52 | 98.64 | 99.52 | 70.36 | 57.05 | 70.71 |
| SCMVC (2024) | _91.20_ | _79.83_ | _91.20_ | 99.38 | 98.36 | 99.38 | 99.66 | 98.99 | 99.66 | 72.36 | 61.45 | 72.71 |
| DDMVC (2025) | 76.20 | 72.17 | 76.20 | 86.20 | 85.48 | 86.37 | 98.41 | 95.52 | 98.41 | 72.64 | 60.75 | 72.64 |
| DMVC_MIC (2025) | 82.60 | 75.19 | 82.60 | 89.10 | 93.67 | 89.10 | 96.24 | 89.47 | 96.62 | 63.64 | 55.21 | 64.64 |
| SparseMVC (2025) | 77.40 | 73.94 | 80.40 | 97.17 | 94.22 | 97.17 | _99.68_ | _99.12_ | _99.68_ | 65.79 | 53.18 | 65.79 |
| CRRC | **95.60** | **87.11** | **95.60** | **99.52** | **98.74** | **99.52** | **99.74** | **99.21** | **99.74** | **74.00** | **62.50** | **74.00** |

| Datasets | Caltech-4V | | | Caltech-5V | | | Animal | | | Cifar100 | | |
|---|---|---|---|---|---|---|---|---|---|---|---|---|
| metrics | ACC | NMI | PUR | ACC | NMI | PUR | ACC | NMI | PUR | ACC | NMI | PUR |
| MFLVC (2022) | 80.71 | 70.03 | 80.71 | 81.86 | 71.25 | 81.86 | 20.65 | 32.44 | 23.03 | 82.68 | 95.60 | 82.68 |
| CVCL (2023) | 76.32 | 67.09 | 76.32 | 77.60 | 69.94 | 77.60 | 33.36 | 46.00 | 38.53 | 61.28 | 87.12 | 64.86 |
| GCFAgg (2023) | 72.71 | 64.62 | 72.71 | 82.50 | 71.23 | 82.50 | 20.26 | 31.92 | 22.45 | 95.97 | 99.35 | 96.05 |
| DCMVC (2023) | 72.86 | 59.72 | 73.79 | 82.43 | 72.69 | 82.43 | _40.65_ | _50.29_ | _48.02_ | 98.97 | 98.95 | 98.97 |
| HFMVC (2024) | 69.21 | 60.59 | 69.21 | 74.79 | 64.49 | 74.79 | 34.31 | 48.99 | 36.72 | 60.74 | 78.21 | 60.29 |
| CSOT (2024) | 75.14 | 67.45 | 75.14 | 79.21 | 68.48 | 79.21 | 21.37 | 35.96 | 27.14 | 87.05 | 97.46 | 87.05 |
| AccMVC (2024) | _81.36_ | _70.17_ | _81.36_ | 88.07 | 79.87 | 88.07 | 29.17 | 41.32 | 33.58 | _99.46_ | 98.43 | _98.81_ |
| SCMVC (2024) | 80.14 | 69.30 | 80.14 | _90.21_ | _81.78_ | _90.21_ | 39.16 | 49.04 | 43.71 | 98.30 | _99.50_ | 98.70 |
| DDMVC (2025) | 76.64 | 69.88 | 76.57 | 88.25 | 80.15 | 88.25 | 24.11 | 36.39 | 28.35 | 72.20 | 86.28 | 76.28 |
| DMVC_MIC (2025) | 75.00 | 70.13 | 75.00 | 84.29 | 72.48 | 84.29 | 39.78 | 50.38 | 47.05 | 95.60 | 98.40 | 96.12 |
| SparseMVC (2025) | 75.93 | 64.10 | 75.93 | 78.64 | 66.03 | 78.64 | 35.00 | 44.02 | 40.15 | 98.26 | 99.35 | 98.71 |
| CRRC | **86.50** | **75.98** | **86.50** | **92.21** | **84.58** | **92.21** | **41.92** | **51.91** | **48.33** | **99.98** | **99.96** | **99.98** |

All input samples are reshaped into vectors. For each view, a fully connected autoencoder (Fc500–Fc500–Fc2000–Fc64) extracts low-level features, followed by a view-specific MLP (Fc64–Fc20) for view-specific representations. A global fusion MLP (Fc($m \times 64$)–Fc256–Fc20) produces $\mathbf{H}$, and the dynamic gating module uses a two-layer fully connected network (Fc40–Fc128–Fc20) with ReLU and sigmoid activations. Experiments run on a Windows system with an Intel i7-12700K CPU, 32GB RAM, and RTX 4070 GPU (12GB). Implementation uses PyTorch 2.3.1, Python 3.8.20, and CUDA 11.8. Training employs Adam (lr=0.0003) with batch size 256. Pre-training lasts 200 epochs using reconstruction loss, followed by 50 epochs of comparison training, dataset-dependent.

## 4.2 COMPARATIVE RESULT ANALYSIS

The clustering results on eight public multi-view datasets are summarized in Table 1. Compared with the second-best method SCMVC, which relies on direct adaptive view weighting and thus suppresses subtle cross-view semantics due to the lack of residual fusion, CRRC better preserves view-specific information while integrating complementary cues. Consequently, CRRC achieves 4.4% and 7.28% gains in ACC and NMI on the NGs dataset AccMVC, which relies on anchor-sharing and cluster-wise contrast by reweighting negative pairs, also lacks fine-grained modeling of cross-view relations. CRRC overcomes this by combining residual recalibration with dynamic gating and attention-based weighting, which adaptively suppress redundant views and emphasize relevant patterns, enabling stronger local-global consistency(e.g., over 5% gains on all metrics on Caltech-4V and a 0.52% ACC improvement on Cifar100). The latest method SparseMVC mitigates cross-view sparsity variation via entropy-matched sparse encoding and reweighting, but its sparsity-driven fusion may underuse complementary semantics when sparsity gaps are small. CRRC's residual-based fine-grained recalibration yields superior results(e.g., a 0.06% ACC gain on MNIST-USPS and substantial improvements on other datasets). CRRC consistently outperforms recent DMVC methods by preserving view information, enhancing complementarity, and recalibrating cross-view relations, achieving robust clustering across datasets.

## 4.3 MODEL ANALYSIS

### 4.3.1 CONVERGENCE ANALYSIS

To verify the convergence of CRRC, we plot the curves of ACC, NMI, and loss over training epochs in Figure 2. The loss decreases rapidly at the beginning and then stabilizes, while ACC and NMI steadily increase before leveling off. These trends confirm that CRRC optimizes effectively and achieves stable, strong performance during training. More experiments are provided in Appendix A.2.1.

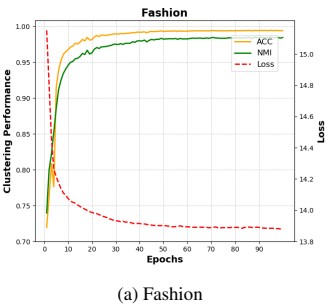 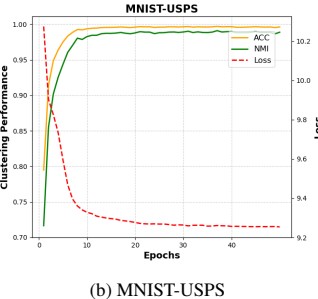

(a) Fashion          (b) MNIST-USPS

Figure 2: Clustering metrics (ACC, NMI) and loss curves over training epochs on the Fashion and MNIST-USPS datasets.

### 4.3.2 PARAMETER SENSITIVITY ANALYSIS

To evaluate the robustness of our method to the temperature parameter $\tau_1$ in Eq. (12), we test it on eight benchmark datasets with $\tau \in \{0.1, 0.3, 0.5, 0.7, 1\}$ (see Figure 3). When $\tau$ is too small (e.g., 0.1), performance drops significantly due to excessive alignment across views, which impairs the preservation of view-specific and structural information. As $\tau$ increases, performance improves and stabilizes between 0.5 and 1, showing that our model is robust to a wide range of $\tau$ values.

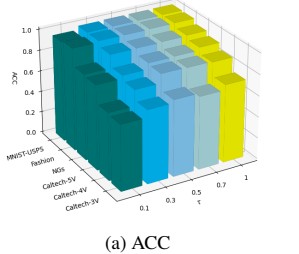 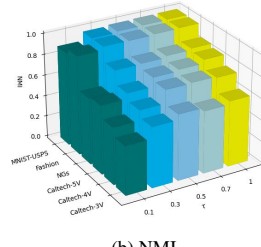 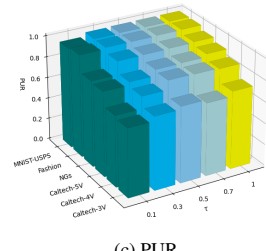

(a) ACC          (b) NMI          (c) PUR

Figure 3: Parameter sensitivity analysis of $\tau$ on NGs, Fashion, MNIST-USPS, and Caltech-XV datasets.

### 4.3.3 VISUALIZATION ANALYSIS OF FEATURE CLUSTERING PROCESS

To further validate the discriminative ability of our model, we utilize t-SNE (Maaten & Hinton, 2008) to visualize the global semantic representation $\mathbf{H}$ at various training stages. As shown in Figure 4, the feature boundaries become increasingly clear as training progresses. Inter-class clusters gradually emerge and evolve into well-separated and compact structures. This progression illustrates that the proposed fusion mechanism and training strategy effectively promote discriminative representations and improve clustering quality. More experiments are provided in Appendix A.2.2.

### 4.3.4 ABLATION STUDY

To assess each component's contribution, we progressively remove modules from CRRC and perform ablations on the Caltech datasets (Table 2); results on other datasets are in Appendix A.2.3.

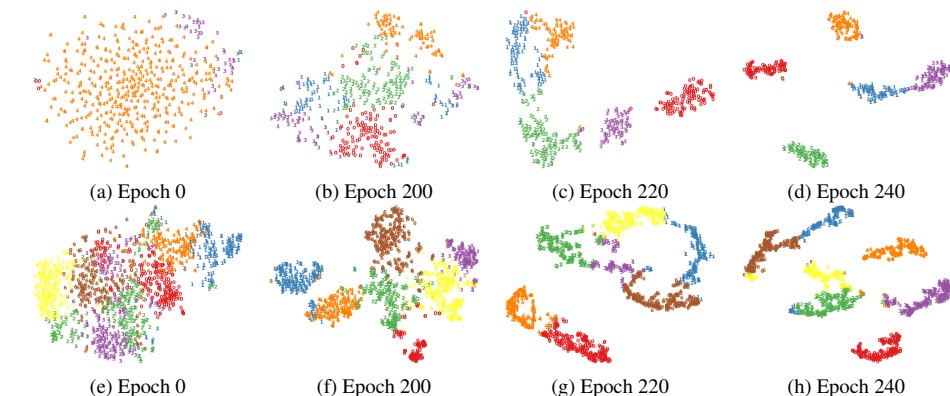

Figure 4: t-SNE visualizations of clustering results on datasets NGs (a)–(d) and Caltech-5V (e)–(h) with increasing training epochs.

Table 2: Ablation results on Caltech-3V, Caltech-4V, and Caltech-5V datasets.

| Model | $L_{\text{rec}}$ | RF | ACW | DGF | Caltech-3V | | | Caltech-4V | | | Caltech-5V | | |
|---|---|---|---|---|---|---|---|---|---|---|---|---|---|
| | | | | | ACC | NMI | PUR | ACC | NMI | PUR | ACC | NMI | PUR |
| $M_1$ | ✓ | | | | 55.79 | 42.83 | 57.29 | 61.43 | 49.73 | 61.43 | 68.50 | 56.31 | 71.64 |
| $M_2$ | ✓ | | ✓ | | 70.79 | 60.58 | 71.36 | 73.71 | 63.17 | 73.71 | 88.86 | 81.23 | 88.86 |
| $M_3$ | ✓ | | | ✓ | 70.71 | 59.36 | 70.71 | 64.29 | 46.91 | 64.29 | 82.57 | 70.18 | 82.57 |
| $M_4$ | ✓ | ✓ | | | 71.64 | 59.16 | 72.07 | 72.00 | 61.60 | 72.00 | 85.43 | 76.21 | 85.43 |
| $M_5$ | ✓ | | ✓ | ✓ | 72.14 | 57.33 | 72.14 | 72.86 | 61.57 | 72.86 | 89.43 | 81.59 | 89.43 |
| $M_6$ | ✓ | ✓ | ✓ | | 67.57 | 55.92 | 67.57 | 78.79 | 70.85 | 78.79 | 86.36 | 78.87 | 86.36 |
| $M_7$ | ✓ | ✓ | | ✓ | 70.93 | 62.00 | 70.93 | 78.71 | 67.14 | 78.71 | 90.71 | 83.17 | 90.71 |
| CRRC | ✓ | ✓ | ✓ | ✓ | **74.00** | **62.50** | **74.00** | **86.50** | **75.98** | **86.50** | **92.21** | **84.58** | **92.21** |

$M_1$, using only the reconstruction loss $L_{\text{rec}}$, achieves 55.79% ACC on Caltech-3V, showing that reconstruction alone cannot capture discriminative clustering structures. $M_2$, $M_3$, and $M_4$ retain only one module at a time, namely the attention-based cross-view weighting (ACW), dynamic gating fusion (DGF), and residual fusion (RF), respectively. Each improves upon $M_1$, but all lag behind the full model, showing that no single module suffices. $M_5$ replaces RF with direct concatenation while keeping ACW and DGF. On Caltech-5V, its ACC decreases by 2.78% compared with CRRC, confirming that residual fusion better preserves complementary information. $M_6$ removes the DGF module while retaining RF and ACW. On Caltech-3V, its ACC and NMI decrease by 6.43% and 6.58%, highlighting the role of dynamic gating in regulating residual flow and enhancing information integration. $M_7$ removes the ACW module while keeping RF and DGF. On Caltech-4V, its ACC decreases by 7.79%, underscoring the necessity of adaptive weighting for modeling cross-view semantic correlations. Overall, CRRC achieves the best performance, demonstrating that RF, ACW, and DGF are indispensable and their combination yields superior clustering.

## 5 CONCLUSION AND LIMITATIONS

This paper presents CRRC, a deep multi-view clustering framework for heterogeneous fusion. CRRC recalibrates cross-view differences via residual connections, preserving view-specific features while capturing complementary semantics. Dynamic gating and attention-based weighting guide integration, and joint optimization of reconstruction and contrastive losses forms a robust semantic space. By jointly leveraging residual recalibration and contrastive learning, CRRC realizes a synergistic co-optimization paradigm, where residual-based view-specific preservation and contrastive objectives are optimized together, yielding superior clustering performance compared to using either module independently. Despite its strong performance, CRRC still depends on some hyperparameters (e.g., hidden size, initialization, activation). Future work may explore lighter or auto-tuned gating to reduce manual tuning. Moreover, the current experiments assume that all views are clean and complete. Future research could investigate robustness to noisy or incomplete views while maintaining effective cross-view integration.

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

# A APPENDIX

## A.1 THEORETICAL ANALYSIS AND PROOFS

### A.1.1 THEORETICAL PROOF OF STABILITY

**Theorem 1.** *Let the residual-fused recalibrated feature representation be $\tilde{\mathbf{R}}^v = g_v \odot \mathbf{R}^{(k \neq v)}$, where $g_v = \sigma(f([\mathbf{R}^v; \mathbf{R}^{(k \neq v)}]))$ is a dynamic gating vector obtained by element-wise sigmoid activation, and $f(\cdot)$ is any differentiable mapping (e.g., the two-layer linear-ReLU-linear transformation in our implementation). Then, for any differentiable loss function $\mathcal{L}$, the gating gradient $\frac{\partial g_v}{\partial \mathbf{R}^v}$ is non-zero under mild non-degeneracy assumptions, thus allowing gradients to propagate during backpropagation.*

*Proof.* By the chain rule,

$$\frac{\partial \mathcal{L}}{\partial \mathbf{R}^v} = \frac{\partial \mathcal{L}}{\partial \tilde{\mathbf{R}}^v} \cdot \frac{\partial \tilde{\mathbf{R}}^v}{\partial g_v} \cdot \frac{\partial g_v}{\partial \mathbf{R}^v}. \tag{15}$$

Given $\tilde{\mathbf{R}}^v = g_v \odot \mathbf{R}^{(k \neq v)}$, we obtain

$$\frac{\partial \tilde{\mathbf{R}}^v}{\partial g_v} = \mathrm{diag}(\mathbf{R}^{(k \neq v)}). \tag{16}$$

Since $f(\cdot)$ is differentiable and $\sigma'(x) = \sigma(x)(1 - \sigma(x)) > 0$ for all finite $x$, it follows that

$$\frac{\partial g_v}{\partial \mathbf{R}^v} = \sigma'(f([\mathbf{R}^v; \mathbf{R}^{(k \neq v)}])) \cdot \frac{\partial f([\mathbf{R}^v; \mathbf{R}^{(k \neq v)}])}{\partial \mathbf{R}^v}. \tag{17}$$

This Jacobian is non-zero under mild non-degeneracy assumptions on $f(\cdot)$ and the inputs (e.g., non-degenerate weight initialization and avoidance of sigmoid saturation), hence the gating path permits effective gradient propagation. Together with the residual shortcut, which provides an identity mapping for direct gradient flow, our dynamic gating structure helps mitigate vanishing gradients and supports stable training in practice. □

### A.1.2 LIPSCHITZ CONTINUITY ANALYSIS

We now prove that the residual modulation function

$$\mathcal{M}(\mathbf{R}^v, \mathbf{R}^{(k \neq v)}) = \sigma(f([\mathbf{R}^v; \mathbf{R}^{(k \neq v)}])) \odot \mathcal{G}(\mathbf{R}^{(k \neq v)}) \tag{18}$$

is Lipschitz continuous, ensuring stable optimization during training.

**Theorem 2.** *There exists a constant $K > 0$ such that for any inputs $\mathbf{R}^v$, $\mathbf{R}^{(k \neq v)}$, $\mathbf{R}^{(k \neq v)'}$, the residual modulation function*

$$\mathcal{M}(\mathbf{R}^v, \mathbf{R}^{(k \neq v)}) = \sigma(f([\mathbf{R}^v; \mathbf{R}^{(k \neq v)}])) \odot \mathcal{G}(\mathbf{R}^{(k \neq v)}) \tag{19}$$

*satisfies the Lipschitz condition*

$$\left\| \mathcal{M}(\mathbf{R}^v, \mathbf{R}^{(k \neq v)}) - \mathcal{M}(\mathbf{R}^{v'}, \mathbf{R}^{(k \neq v)'}) \right\|$$
$$\leq K \left( \left\| \mathbf{R}^v - \mathbf{R}^{v'} \right\| + \left\| \mathbf{R}^{(k \neq v)} - \mathbf{R}^{(k \neq v)'} \right\| \right). \tag{20}$$

*Proof.* Define

$$g = \sigma(f([\mathbf{R}^v; \mathbf{R}^{(k \neq v)}])),$$
$$g' = \sigma(f([\mathbf{R}^{v'}; \mathbf{R}^{(k \neq v)'}])). \tag{21}$$

Since $\sigma$ is $\frac{1}{4}$-Lipschitz continuous and $f(\cdot)$ is differentiable on a bounded domain, there exists a constant $L_f > 0$ such that

$$\|g - g'\| \leq \frac{1}{4} L_f \left( \|\mathbf{R}^v - \mathbf{R}^{v'}\| + \|\mathbf{R}^{(k \neq v)} - \mathbf{R}^{(k \neq v)'}\| \right). \tag{22}$$

The fusion outputs are

$$\mathcal{M}(\mathbf{R}^v, \mathbf{R}^{(k \neq v)}) = g \odot \mathcal{G}(\mathbf{R}^{(k \neq v)}),$$
$$\mathcal{M}(\mathbf{R}^{v'}, \mathbf{R}^{(k \neq v)'}) = g' \odot \mathcal{G}(\mathbf{R}^{(k \neq v)'}). \tag{23}$$

Using the triangle inequality and Hadamard product properties, we get

$$\left\| g \odot \mathcal{G}(\mathbf{R}^{(k \neq v)}) - g' \odot \mathcal{G}(\mathbf{R}^{(k \neq v)'}) \right\|$$
$$\leq \|g - g'\| \cdot \left\| \mathcal{G}(\mathbf{R}^{(k \neq v)}) \right\|_{\max} + \|g'\|_{\max} \cdot \left\| \mathcal{G}(\mathbf{R}^{(k \neq v)}) - \mathcal{G}(\mathbf{R}^{(k \neq v)'}) \right\|. \tag{24}$$

Assuming $\mathcal{G}$ is Lipschitz continuous with constant $L_{\mathcal{G}}$, combining inequalities yields

$$\left\| \mathcal{M}(\mathbf{R}^v, \mathbf{R}^{(k \neq v)}) - \mathcal{M}(\mathbf{R}^{v'}, \mathbf{R}^{(k \neq v)'}) \right\|$$
$$\leq \frac{1}{4} L_f \left( \|\mathbf{R}^v - \mathbf{R}^{v'}\| + \|\mathbf{R}^{(k \neq v)} - \mathbf{R}^{(k \neq v)'}\| \right) \left\| \mathcal{G}(\mathbf{R}^{(k \neq v)}) \right\|_{\max} \tag{25}$$
$$+ \|g'\|_{\max} L_{\mathcal{G}} \left\| \mathbf{R}^{(k \neq v)} - \mathbf{R}^{(k \neq v)'} \right\|.$$

Hence, the residual modulation function is Lipschitz continuous with constant

$$K = \frac{1}{4} L_f \cdot \left\| \mathcal{G}(\mathbf{R}^{(k \neq v)}) \right\|_{\max} + \|g'\|_{\max} L_{\mathcal{G}}. \tag{26}$$

This completes the proof. $\square$

### A.1.3 CONVERGENCE ANALYSIS

We analyze the convergence behavior of the optimization problem defined by our fusion function:

$$\min_{\theta, f} \mathbb{E}_{(\mathbf{R}^v, \mathbf{R}^{(k \neq v)})} \left[ \left\| \mathcal{M}(\mathbf{R}^v, \mathbf{R}^{(k \neq v)}; \theta, f) - y \right\|^2 \right], \tag{27}$$

where $f(\cdot)$ is any differentiable mapping, e.g., the two-layer linear-ReLU-linear transformation in our implementation.

**Theorem 3** (Convergence Rate of Fusion Optimization). *Assume that the loss function $\mathcal{L}(\theta)$ is smooth and strongly convex, and the stochastic gradient satisfies bounded variance:*

$$\mathbb{E}\left[ \left\| \nabla \mathcal{L}(\theta) - \nabla \hat{\mathcal{L}}(\theta) \right\|^2 \right] \leq \sigma^2. \tag{28}$$

*Let $\theta_T$ be the model parameters after $T$ steps of stochastic gradient descent (SGD) with learning rate $\eta$. Then the expected optimization error satisfies:*

$$\mathbb{E}\left[ \mathcal{L}(\theta_T) - \mathcal{L}^* \right] \leq \frac{\|\theta_0 - \theta^*\|^2}{2\eta T} + \frac{\eta \sigma^2}{2} + \mathcal{O}\left( L_f^4 / T \right), \tag{29}$$

*where $L_f$ is the Lipschitz constant of the mapping $f(\cdot)$.*

*Proof.* The result follows from classical convergence theory of SGD for smooth and strongly convex functions. Given the update rule:

$$\theta_{t+1} = \theta_t - \eta \nabla \hat{\mathcal{L}}(\theta_t), \tag{30}$$

it is well known that:

$$\mathbb{E}\left[ \mathcal{L}(\theta_T) - \mathcal{L}^* \right] \leq \frac{\|\theta_0 - \theta^*\|^2}{2\eta T} + \frac{\eta \sigma^2}{2}. \tag{31}$$

In our setting, the fusion function $\mathcal{M}(\mathbf{R}^v, \mathbf{R}^{(k \neq v)}; \theta, f)$ incorporates a general differentiable gating mechanism $f(\cdot)$. From Theorem 2, the fusion function is Lipschitz continuous with constant $L_f$, which affects the smoothness and stability of the optimization.

Using gradient perturbation analysis, this results in an additional error term on the order of $L_f^4 / T$. Thus, the total expected optimization error is bounded as in equation 29. When the learning rate is chosen as $\eta = \mathcal{O}(1/\sqrt{T})$, the expected optimization error decays at the rate $\mathcal{O}(1/\sqrt{T})$, consistent with standard SGD convergence behavior.

This completes the proof. $\square$

## A.2 ADDITIONAL EXPERIMENTAL ANALYSIS

### A.2.1 CONVERGENCE ANALYSIS

To verify the convergence of CRRC, we plot the curves of ACC, NMI, and loss over training epochs in Figure 5. The loss decreases rapidly at the beginning and then stabilizes, while ACC and NMI steadily increase before leveling off. These trends confirm that CRRC optimizes effectively and achieves stable, strong performance during training.

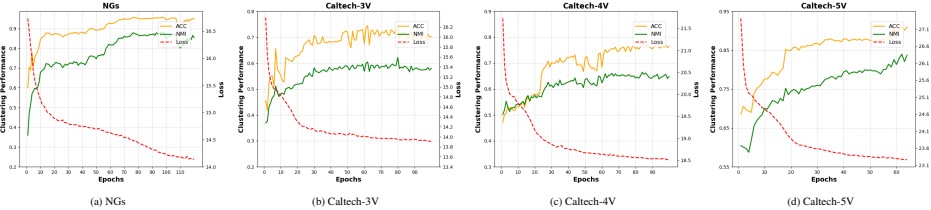

| (a) NGs | (b) Caltech-3V | (c) Caltech-4V | (d) Caltech-5V |

Figure 5: Clustering metrics (ACC, NMI) and loss curves over training epochs on the NGs, Caltech-3V, Caltech-4V and Caltech-5V datasets.

### A.2.2 VISUALIZATION ANALYSIS OF FEATURE CLUSTERING PROCESS

To further demonstrate the effectiveness of our model in enhancing feature discriminability, we utilize t-SNE Maaten & Hinton (2008) to visualize the global feature representation $\mathbf{H}$ at various training stages. As shown in Figure 6, the feature boundaries become increasingly clear as training progresses. Inter-class clusters gradually emerge and evolve into well-separated and compact structures. This progression illustrates that the proposed fusion mechanism and training strategy effectively promote discriminative representations and improve clustering quality.

### A.2.3 ABLATION STUDY

We further conduct ablation studies on NGs, Fashion, and MNIST-USPS datasets, as reported in Table 3. $M_1$, which uses only $L_{rec}$, performs poorly (e.g., 56.00% ACC on NGs), indicating that reconstruction alone fails to capture discriminative structures. $M_2$, $M_3$, and $M_4$ retain only one module each (ACW, DGF, and RF, respectively). While they improve over $M_1$, their performance is consistently inferior to the full model, showing that a single component is insufficient. $M_5$ replaces RF with direct concatenation while keeping ACW and DGF. On NGs, its ACC drops by 1.00% compared with CRRC, verifying the necessity of residual fusion in preserving complementary semantics. $M_6$ removes DGF while retaining RF and ACW. On NGs, its ACC decreases by 1.20%, suggesting that dynamic gating plays a role in regulating residual flow and enhancing information integration. $M_7$ removes ACW while keeping RF and DGF. On NGs, its NMI decreases by 1.99%, underscoring the importance of adaptive cross-view weighting. Overall, CRRC achieves the best performance across all datasets, demonstrating that RF, ACW, and DGF are indispensable, and their synergy yields superior clustering.

Table 3: Ablation results on NGs, Fashion, and MNIST-USPS datasets.

| Model | $L_{rec}$ | RF | ACW | DGF | NGs | | | Fashion | | | MNIST-USPS | | |
|---|---|---|---|---|---|---|---|---|---|---|---|---|---|
| | | | | | ACC | NMI | PUR | ACC | NMI | PUR | ACC | NMI | PUR |
| $M_1$ | ✓ | | | | 56.00 | 38.83 | 57.00 | 67.60 | 66.03 | 72.21 | 54.72 | 43.67 | 54.72 |
| $M_2$ | ✓ | | ✓ | | 93.80 | 83.22 | 93.80 | 99.06 | 97.65 | 99.06 | 99.46 | 98.45 | 99.46 |
| $M_3$ | ✓ | | | ✓ | 94.00 | 82.52 | 94.00 | 99.00 | 97.53 | 99.00 | 99.44 | 98.38 | 99.44 |
| $M_4$ | ✓ | ✓ | | | 94.60 | 85.05 | 94.60 | 99.46 | 98.53 | 99.46 | 99.60 | 98.90 | 99.60 |
| $M_5$ | ✓ | | ✓ | ✓ | 94.60 | 85.10 | 94.60 | 98.90 | 97.36 | 98.90 | 99.44 | 98.37 | 99.44 |
| $M_6$ | ✓ | ✓ | ✓ | | 94.40 | 84.04 | 94.40 | 99.50 | 98.70 | 99.50 | 99.66 | 99.00 | 99.66 |
| $M_7$ | ✓ | ✓ | | ✓ | 94.60 | 85.12 | 94.40 | 99.47 | 98.61 | 99.47 | 99.64 | 98.94 | 99.64 |
| CRRC | ✓ | ✓ | ✓ | ✓ | **95.60** | **87.11** | **95.60** | **99.52** | **98.74** | **99.52** | **99.74** | **99.21** | **99.74** |

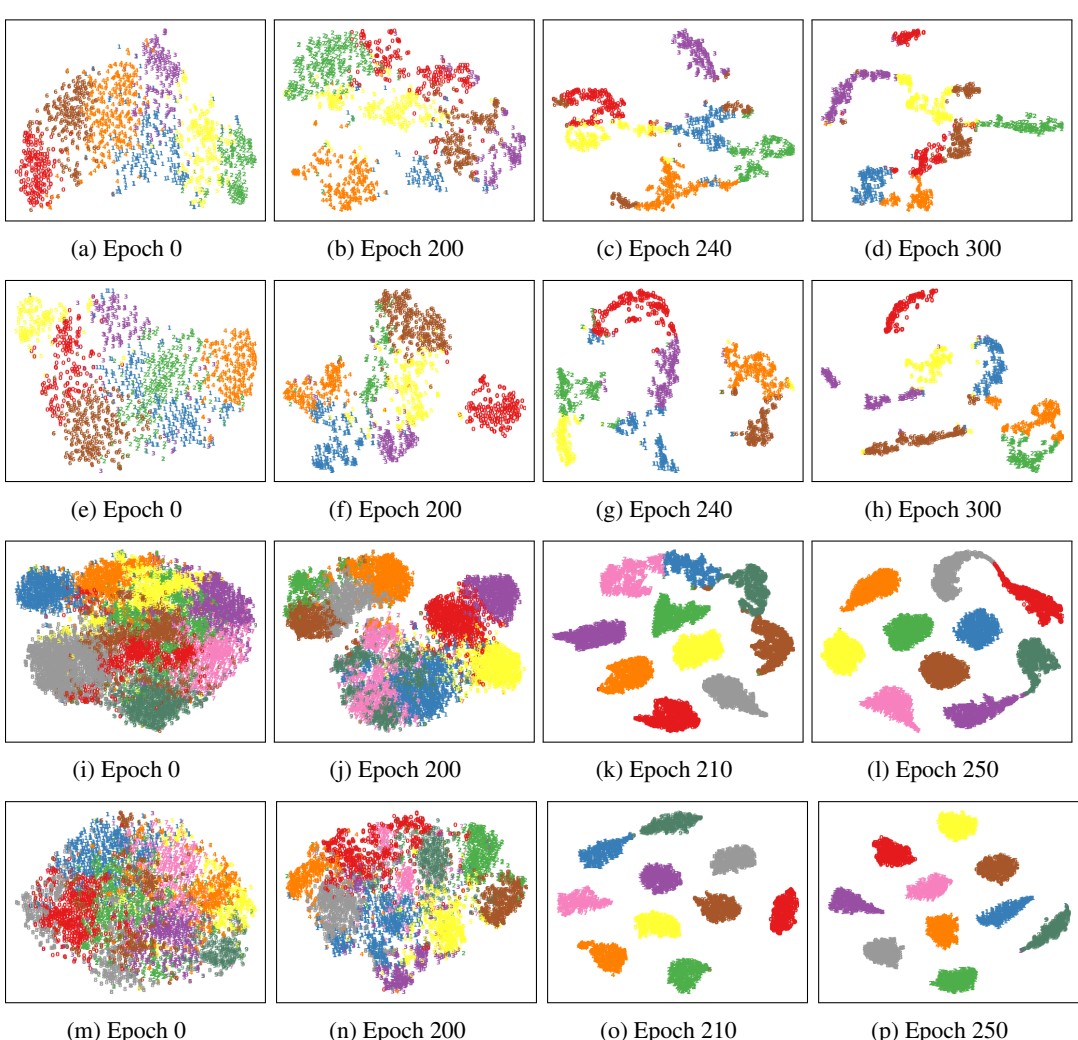

(a) Epoch 0    (b) Epoch 200    (c) Epoch 240    (d) Epoch 300

(e) Epoch 0    (f) Epoch 200    (g) Epoch 240    (h) Epoch 300

(i) Epoch 0    (j) Epoch 200    (k) Epoch 210    (l) Epoch 250

(m) Epoch 0    (n) Epoch 200    (o) Epoch 210    (p) Epoch 250

Figure 6: t-SNE visualizations of clustering results on the Caltech-3V, Caltech-4V, Fashion and MNIST-USPS datasets with increasing training epochs. (a)–(d) show the results on the Caltech-3V dataset. (e)–(h) show the results on the Caltech-4V dataset. (i)–(l) show the results on the Fashion dataset. (m)–(p) show the results on the MNIST-USPS dataset.

## A.3 USE OF LARGE LANGUAGE MODELS

We only used Large Language Models (LLMs) for language polishing, such as improving the grammar, readability, and clarity of the paper. No part of the research ideas, technical content, experimental design, or results was generated by LLMs.

