# OpenReview forum: "CRRC: Residual Cross-view Learning for Deep Multi-view Clustering"
_ICLR.cc/2026/Conference — Submitted to ICLR 2026_

### Official Review · Reviewer_ix5q · 2025-10-25

**Soundness:** 3
**Presentation:** 3
**Contribution:** 3
**Rating:** 6
**Confidence:** 4

**Summary:**

This paper presents the CRRC framework, which effectively addresses the balance between shared semantics and view-specific heterogeneity in multi-view clustering by incorporating cross-view residual recalibration, dynamic gating fusion, and attention-based weighting mechanisms. The CRRC framework fuses cross-view information through residual connections, preserving view-specific differences, while leveraging dynamic adjustments and attention mechanisms to enhance feature discriminability and consistency. Experimental results demonstrate that CRRC outperforms existing methods on several benchmark datasets, showing superior performance in ACC, NMI, and PRI metrics.

**Strengths:**

1.The CRRC framework combines residual recalibration with contrastive learning, proposing a clustering method that is highly adaptive and robust in the presence of noise and heterogeneity. This method is not only original in theory but also highly significant in practical applications.

2.The introduction of the cross-view residual recalibration mechanism effectively addresses the information loss caused by direct concatenation or alignment of views, allowing for the flexible integration of complementary information from different views.

3.Through the Dynamic Gating Fusion module, the framework utilizes an attention mechanism to adaptively adjust the degree of fusion of cross-view information.

4.The paper extensively validates the CRRC framework’s superiority on multiple benchmark datasets, showing that CRRC outperforms existing methods in terms of clustering accuracy (ACC), normalized mutual information (NMI), and purity (PRI).

**Weaknesses:**

1.In Figure 1, the framework lacks a detailed description of the overall process flow.

2.The paper incorporates contrastive learning as part of the overall framework, but its independent contribution has not been sufficiently quantified.

3.The model employs only reconstruction and contrastive losses, which enhance feature discriminability and consistency but do not include clustering-oriented losses such as clustering loss or cluster-level contrastive loss. This may result in insufficiently targeted optimization for clustering objectives.

4.Although the Dynamic Gating Fusion (DGF) module helps regulate cross-view information flow, insufficient or overly restrictive learning of gating weights may lead to the excessive suppression of useful information.

**Questions:**

1.Although the contrastive loss and reconstruction loss help improve model performance, they are not the core innovation of this paper. Why were these two losses specifically chosen? Are there other loss functions that could further enhance the effectiveness of the residual recalibration and dynamic gating fusion mechanisms?

2.The paper employs contrastive learning as part of its loss function, but has the performance of the framework been evaluated without it? Do the ablation experiments demonstrate the specific contribution of contrastive learning to multi-view clustering? If the contrastive module were removed, could other components (such as DGF and RF) still effectively enhance clustering performance, or does contrastive learning play an indispensable role in the framework?

3.This paper uses a self-attention mechanism to weight cross-view information during fusion, a strategy also seen in other methods. What distinguishes the application of self-attention in this work from its use in previous approaches?

4.In the related work section, the authors mention that multi-view clustering typically follows two training paradigms: end-to-end and two-stage. Which paradigm does this paper adopt, and what are the reasons behind this choice? Compared with the alternative, does the selected approach offer advantages in optimization efficiency or training stability? Additionally, has the time complexity of the model been evaluated?

---

> ### Author Response · Authors · 2025-11-21
> **Reply to Reviewer ix5q (Part 1 of 2)**
>
> Thanks for your careful review. We are glad to address your questions one by one.
>
> **W1:** We will add a new subsection (e.g., 3.1 Overall Workflow) at the beginning of Section 3 to give a concise step-by-step overview:
> "As shown in Figure 1, CRRC follows an end-to-end workflow with three main stages:
>
> (1) **Multi-view Encoding and Reconstruction**: Each view $\mathbf{X}^v$ is encoded into a latent feature $\mathbf{Z}^v$ by its autoencoder while being reconstructed to ensure feature fidelity.
>
> (2) **Global and Local Representation Generation**: All $\mathbf{Z}^v$ are concatenated and passed through a global MLP to form the unified semantic representation $\mathbf{H}$. Each $\mathbf{Z}^v$ is also fed into a view-specific MLP to obtain the local representation $\mathbf{R}^v$.
>
> (3) **Residual Recalibration**: For each view $v$, recalibration includes:
> * a. Attention Weighting: Using $\mathbf{H}$ as query, attention weights $\alpha^{k\neq v}$ are computed over other views' $\mathbf{R}^{k\neq v}$, producing weighted complementary information.
> * b. Dynamic Gating: A gating vector $\mathbf{g}_v$ is generated from $\mathbf{R}^v$ and the cross-view information.
> * c. Residual Fusion: The gated complementary information is fused with $\mathbf{R}^v$ to obtain the recalibrated feature $\hat{\mathbf{R}}^v$."
>
> (4) **Contrastive Learning and Clustering**:
> The global representation $\mathbf{H}$ and all recalibrated features $\{\hat{\mathbf{R}}^v\}$ are used for cross-view contrastive learning, pulling together different views of the same instance while pushing apart views of different instances. Finally, $\mathbf{H}$ is used for k-means clustering to output the cluster assignments $\mathbf{Y}$."
>
> ---
> **W2**: Our framework aims to **learn a representation space** that is both discriminative and cross-view consistent. The Residual Recalibration module builds this representation by preserving view-specific information while injecting controlled complementary cues, producing richer features $\hat{\mathbf{R}}^v$ and $\mathbf{H}$. The Contrastive Learning module then structures this space by enforcing a clear objective—pulling positives together and separating negatives—ensuring the learned features form a coherent semantic space. Our ablation already supports this: in Section 4.3.4, the baseline $M_1$ (only using $L_{rec}$) performs significantly worse, demonstrating that the contrastive semantic consistency loss $L_{cm}$ is essential for forming effective cluster structures. Following your suggestion, we will refine the Introduction and Conclusion to clearly highlight our key contribution: **a novel residual recalibration + contrastive learning co-optimization paradigm**, whose superiority over alternative objectives or single-module variants is empirically validated, rather than merely adding a residual component.
>
> ---
> **W3**: Our key idea is that once a semantically consistent and highly discriminative shared feature space is learned, simple clustering methods (e.g., k-means) become naturally optimal. Thus, our framework prioritizes building this space rather than imposing clustering constraints during training. Explicit clustering losses (e.g., KL divergence) rely on dynamically generated pseudo-labels that are often **inaccurate and unstable** in early training, leading to trivial or suboptimal solutions. We avoid this risk by using more stable objectives to guide representation learning.
> By learning features that are inherently cluster-friendly instead of directly manipulating cluster assignments, we achieve clustering in a more robust and empirically superior way. We will revise the methodology section to clarify this design philosophy.
>
> ---
> **W4**: We use the Sigmoid function as a soft gating mechanism, producing continuous values in (0,1). This enables fine-grained control **rather than binary decisions**, allowing the model to partially pass cross-view information instead of fully blocking it. Even when suppression occurs, the gate typically learns small non-zero values, preventing complete information cutoff.
> Moreover, the DGF module receives complementary information already filtered by the ACW module, which performs coarse relevance weighting. As a result, the input to DGF is cleaner and more reliable, allowing DGF to focus on fine-level adjustment rather than identifying noise, thereby reducing the risk of excessive suppression. A strong safety guarantee is provided by the residual connection: $\hat{\mathbf{R}}^v = \mathbf{R}^v + \mathbf{g}_v \odot G(\dots)$. The original view-specific feature $\mathbf{R}^v$ is always preserved. Even if the gated residual becomes negligible, the output simply falls back to $\mathbf{R}^v$, ensuring no loss of original information and establishing a robust lower bound.
> Together, these mechanisms make DGF module resistant to **"excessive suppression"**, and experiments confirm its practical effectiveness. We will include a discussion of this trade-off in the revised manuscript.

---

> ### Author Response · Authors · 2025-11-21
> **Reply to Reviewer ix5q (Part 2 of 2)**
>
> **Q1**: **$L_{rec}$** ensures reliability rather than novelty, acting as a strong unsupervised regularizer that guarantees each encoder captures clean, complete view-specific features. Residual recalibration depends on $\mathbf{Z}^v$ and $\mathbf{R}^v$, so noisy or weak features would compromise the mechanism.
> **$L_{cm}$** drives recalibration toward clustering by enforcing a representation space that is consistent across views and discriminative across instances. It aligns different views of the same instance ($\mathbf{H}_i$ and $\hat{\mathbf{R}}_i^v$) and separates different instances ($\mathbf{H}_i$ and $\hat{\mathbf{R}}_j^v$, $j\neq i$), preventing collapse and ensuring a cluster-friendly space.
> Together, residual recalibration with these two basic losses suffices for strong clustering; adding complex losses (e.g., KL divergence) may introduce extra hyperparameters, instability, and noisy pseudo-labels, which could obscure the core contribution.
>
> **Q2**: Contrastive learning itself is a "generic objective", while our contribution lies in a novel "feature generator" (RF + DGF) that makes it far more effective.
> **Ablation study (model $M_1$) confirms this**: removing the contrastive loss causes ACC on Caltech-3V to drop by 18.21%, with similar declines on other datasets. This shows that without the semantic-consistency signal from contrastive learning, the model fails to form meaningful clusters, even with full RF and DGF.
> Crucially, CRRC enhances contrastive learning by refining and aligning features through RF and DGF before applying the loss, removing noise and cross-view inconsistencies. While other losses (e.g., clustering losses) could be used, contrastive learning aligns naturally with instance-level consistency and provides stable training. The fact that our model surpasses other contrastive-based methods (DCMVC, AccMVC) shows that the gains stem mainly **from our feature generator, not merely the contrastive objective**.
>
> **Q3**: Our key contribution is redefining attention within a residual recalibration framework. Unlike existing multi-view clustering methods where attention directly produces the final fused representation, our ACW's sole purpose is to compute weights for the "residual term". It evaluates the importance of $\mathbf{R}^{(k \neq v)}$ for recalibrating view $v$, producing a weighted complementary vector further refined by DGF before being added to the original features.
> ACW and DGF form a two-stage regulation mechanism:
> * **Stage 1**: Attention at the view level selects "which views" contribute more complementary information, acting as a coarse filter.
> * **Stage 2**: Dynamic gating at the feature level adjusts "how much" of this information to integrate per feature dimension, acting as a fine mixer.
>
> **Q4**: The proposed CRRC framework adopts a **"Pre-training + Joint Fine-tuning" end-to-end paradigm**. Each view’s autoencoder is first pre-trained independently, then the whole system—including autoencoders, residual recalibration, and contrastive loss—is jointly optimized.
> This hybrid approach combines the benefits of two-stage and end-to-end paradigms:
> - Pre-training provides stable, meaningful features, reducing task misalignment risk in traditional two-stage methods.
> - Joint fine-tuning avoids the instability of training all modules from scratch, common in end-to-end models with autoencoders and contrastive learning.
>
> **Compared with pure end-to-end training**, this strategy ensures better stability and faster convergence. Gating and attention modules receive informative inputs from the start, preventing early-stage failures. Figure 2 shows that loss and ACC/NMI curves stabilize quickly. Pre-training adds epochs but converges rapidly using only MSE loss, easing overall training.
>
> ---
> We analyzed the time complexity and will include a detailed discussion in the revised manuscript. The main computational cost of CRRC comes from the following components, determined by the number of views $m$ and samples $n$:
> * **Autoencoder Forward Pass:** $\mathcal{O}(m \cdot n \cdot D_v \cdot d)$, where $D_v$ is the input dimension of view $v$ and $d$ is the latent dimension.
> * **Global Representation $\mathbf{H}$ Generation:** $\mathcal{O}(n \cdot (m \cdot d) \cdot d_h)$, where $d_h$ is the hidden dimension of the global MLP.
> * **Attention Module:** Computes similarity between $\mathbf{H}$ and $\mathbf{R}^k$ from the other $(m-1)$ views, with complexity $\mathcal{O}(m^2 \cdot n \cdot d^2)$. Since $m$ is usually small (2-5), this cost is manageable.
> * **Dynamic Gating Module:** A two-layer MLP with complexity $\mathcal{O}(n \cdot m \cdot d \cdot h_g)$, where $h_g$ is the hidden dimension of the gating network.
> * **Contrastive Loss:** Computes pairwise similarities for all samples with complexity $\mathcal{O}(m \cdot n^2 \cdot d)$. Using mini-batches reduces practical cost to $\mathcal{O}(m \cdot B^2 \cdot d)$ ($B=256$ in our experiments), which is standard in contrastive learning.

---

> > ### Comment · Reviewer_ix5q · 2025-11-25
> >
> > Thank you to the authors for their detailed reply, which has resolved some of my concerns. I have decided to maintain my original score, although I would not object if the paper were ultimately rejected.

---

> > > ### Author Response · Authors · 2025-11-27
> > > **Reply to Reviewer ix5q**
> > >
> > > Thank you very much for your response and for letting us know that some of your concerns have been resolved by our rebuttal. If there are any remaining concerns that we may have overlooked or that you feel still require clarification, we would be very grateful if you could kindly point them out. We would be more than happy to provide additional explanations or supporting evidence.

---

### Official Review · Reviewer_hx8C · 2025-10-26

**Soundness:** 3
**Presentation:** 3
**Contribution:** 3
**Rating:** 6
**Confidence:** 4

**Summary:**

This paper addresses the problem of deep multi-view clustering, which aims to integrate complementary information from multiple heterogeneous views to improve clustering performance while balancing shared semantics and view-specific heterogeneity. The proposed CRRC leverages residual connections to recalibrate view-specific features through a dynamic gating fusion module and an attention-based cross-view weighting mechanism. The core contributions include a novel fusion strategy that adaptively incorporates cross-view information while preserving view-specific traits. Experimental results on benchmark datasets demonstrate improved performance.

**Strengths:**

1.	The dynamic gating fusion module and attention-based cross-view weighting provide a novel approach to multi-view fusion by adaptively controlling residual flow
2.	The paper is well-organized, with logical flow from problem statement to method description.
3.	The framework addresses key limitations in multi-view clustering, such as view imbalance and semantic alignment.

**Weaknesses:**

1.	The formulation of dynamic gating mechanisms is quite similar to existing gating architectures in neural networks, such as LSTM gates. The manuscript does not sufficiently distinguish its gating approach from these established methods and justify its fundamental novelty.
2.	The attention mechanism relies on global representation $H$ as queries, but $H$ itself is generated through simple concatenation and MLP processing. This approach may not adequately capture fine-grained cross-view semantic relationships.
3.	The framework heavily relies on existing contrastive learning. The manuscript does not clearly establish how the residual recalibration component provides fundamental advantages over prior methods.

**Questions:**

See weaknesses.

---

> ### Author Response · Authors · 2025-11-21
> **Reply to Reviewer hx8C (Part 1 of 2)**
>
> Thanks for your careful review. We are glad to address your questions one by one.
>
> **W1**: Thank you for your insightful comment regarding the similarity between our dynamic gating mechanism and existing gating architectures such as those in LSTMs. Our approach differs significantly in objective, design, and context:
> * **Different Objectives**:
> LSTM gates manage temporal memory and handle sequential dependencies. Our gating mechanism is designed for multi-view clustering, where it adaptively modulates the residual flow of complementary information from other views to enhance cross-view semantic consistency while preserving view-specific characteristics.
> * **Structural Differences**:
> We do not use multiple gates (input, forget, output) as in LSTMs. Instead, we employ a single gating vector generated dynamically from the concatenation of the current view and other views’ features, specifically to recalibrate residual signals. This gating mechanism is integrated with residual connections and an attention-based weighting module, forming a novel residual cross-view recalibration framework for multi-view clustering.
> * **Novelty**:
> Our contribution is not merely the gating mechanism itself, but the novel paradigm of residual cross-view learning, where gating is used to adaptively fuse complementary information in a way that has not been systematically explored in multi-view clustering before.
>
> ---
> **W2**: Thank you for raising this important point regarding our use of the globally generated representation $\mathbf{H}$ as the query in the attention mechanism.
> The design goal of $\mathbf{H}$ is not to be a perfectly fine-grained representation itself, but to serve as a **unified "semantic anchor"** or "consensus center". By aggregating information from all views, it aims to capture the overall semantic structure of a sample. Using $\mathbf{H}$ as the query allows the view-specific features of each view to be aligned and compared against this common reference point. This provides a stable coordinate system for all views, facilitating robust learning of cross-view consistency and mitigating potential instability or noise from direct pairwise view alignment.
>
> We agree that in isolation, this attention module might have limitations in capturing ultra-fine-grained relationships. However, in our CRRC framework, the attention module works synergistically with the dynamic gating module.
> The Attention-based Cross-view Weighting (ACW) module operates at a macro level, assessing the overall semantic relevance of other views to the current one ("**which views are more important?**").
> The Dynamic Gating Fusion (DGF) module then performs fine-grained, element-wise modulation ("**how much complementary information should be incorporated at each feature dimension?**").
> These two modules are complementary: the attention provides a weighted source, and the gating executes a refined fusion. This collaborative design mitigates the potential coarseness of using a global query alone.
>
> Our ablation studies (Tables 2 and 3 in the manuscript) demonstrate that removing the attention module (M$_7$) leads to a significant performance drop across all datasets. This empirically validates that the attention mechanism built upon $\mathbf{H}$ is crucial and effective for the final performance, despite its straightforward generation process.

---

> ### Author Response · Authors · 2025-11-21
> **Reply to Reviewer hx8C (Part 2 of 2)**
>
> **W3**: Our key innovation, however, **lies not in proposing a new contrastive loss, but in introducing a novel feature representation generation framework** that fundamentally addresses core challenges in multi-view fusion, thereby enabling contrastive learning to perform more effectively.
>
> Most prior contrastive methods (e.g., CVCL[1], DCMVC[2], DDMVC[3]) primarily focus on designing more sophisticated contrastive losses to align given, static feature representations. They treat the features as fixed and strive to "pull" them together.
> Our CRRC framework effects a paradigm shift: we dynamically and adaptively recalibrate and enhance the features **before they are used in contrastive learning**. The residual recalibration module is not a separate pre-processing step but a co-designed core component integrated with the contrastive objective. It produces $\hat{\mathbf{R}}^v$, a representation specifically "tailored" for more effective and robust cross-view contrast.
>
> Traditional fusion methods (e.g., concatenation, weighted average) face a fundamental dilemma: introducing complementary information often dilutes or contaminates the original view-specific information. This forces contrastive learning to operate on "noisy" or "blurred" representations, limiting performance. Our residual connection design structurally resolves this dilemma.
> **The identity path $\mathbf{R}^v$** ensures view-specific information is preserved without loss, forming the basis for discriminability.
> **The residual path $g_v \odot \mathcal{G}(\cdot)$** purely and controllably injects "**incremental information**" from other views.
> This design ensures information is additive rather than substitutive, providing contrastive learning with cleaner, higher-information-density features at the source.
>
> **Our ablation studies (Tables 2 and 3) strongly support the above points**. The significant performance drop of model M$_5$ (which replaces residual fusion with direct concatenation) across all datasets directly proves that the residual recalibration structure itself—beyond just gating or attention—provides a fundamental performance gain.
> It demonstrates that our method addresses a core problem that traditional fusion handles suboptimally.
> In summary, our core contribution is the proposal of "**Residual Cross-view Recalibration**" as a new paradigm for multi-view representation learning. Contrastive learning serves as a powerful and necessary objective function within our framework, but the fundamental driver of performance gain is our proposed feature generation mechanism. We will revise the relevant sections (especially the Introduction and Discussion) in the manuscript to more sharply articulate this paradigm shift and fundamental advantage, and to more clearly differentiate our work from methods that merely apply or tweak contrastive losses.
>
> [1] Deep Multiview Clustering by Contrasting Cluster Assignments. ICCV. 2023
>
> [2] Dual Contrast-Driven Deep Multi-View Clustering. IEEE TRANSACTIONS ON IMAGE PROCESSING. 2024
>
> [3] Deep multi-view clustering with diverse and discriminative feature learning. Pattern Recognition. 2025

---

> > ### Comment · Reviewer_hx8C · 2025-11-24
> >
> > Thanks very much for your detailed replies and these have resolved my problems. Considering the inputs from other reviewers, I would keep my rating as an borderline accept.

---

### Official Review · Reviewer_5zKm · 2025-10-30

**Soundness:** 2
**Presentation:** 3
**Contribution:** 2
**Rating:** 4
**Confidence:** 4

**Summary:**

This paper proposes CRRC to address critical limitations in deep multi-view clustering. The framework integrates three key innovations: residual recalibration (RF) to capture cross-view complementary patterns via residual connections, dynamic gating fusion (DGF) to adaptively adjust residual flow strength, and attention-based cross-view weighting (ACW) to suppress redundant views through semantic correlation scoring. Extensive experiments on benchmarks datasets demonstrate that CRRC outperforms state-of-the art methods.

**Strengths:**

1.	The proposed framework innovatively combines residual learning and attention mechanisms to model cross-view complementarity while preserving view-specific features.
2.	This method achieves state-of-the-art performance across diverse datasets with consistent gains over 9 comparative methods.

**Weaknesses:**

1.	The combination of dynamic gating (DGF), and attention weighting (ACW) may introduce redundant computations. ACW and DGF both aim to suppress irrelevant cross-view signals, potentially leading to overlapping functionality.
2.	While the paper acknowledges sensitivity to the temperature parameter $\tau$, other critical hyperparameters are not systematically evaluated.
3.	The paper assumes clean and complete views, but real-world multi-view data often suffer from noise or missing views. CRRC’s performance under such scenarios is untested.

**Questions:**

Please refer to the Weaknesses.

---

> ### Author Response · Authors · 2025-11-21
> **Reply to Reviewer 5zKm (Part 1 of 2)**
>
> Thanks for your careful review. We are glad to address your questions one by one.
>
> **W1**: Thank you for this insightful and constructive comment. You have rightly highlighted the importance of ensuring clear delineation between modules.
> However, in CRRC's design, the DGF and ACW are meticulously engineered as **complementary modules** that operate at different levels and with distinct responsibilities. Their synergy is crucial for achieving fine-grained fusion, rather than representing functional overlap.
> To use an analogy: if cross-view fusion is like adjusting a shower's water temperature:
> The ACW module acts as the water source selector, deciding whether to mix more "hot water" (View A) or "cold water" (View B);
> The DGF module acts as the main control valve, determining the total flow rate needed from other sources (regardless of temperature), rather than fully closing or opening the current view's tap.
>
> ACW operates at the **"Inter-View" level**: It serves as a fine-grained "signal selector" working at the view granularity. Its core question is: "Among all complementary views, which one(s) are most relevant for enhancing the current view $v$?" It computes attention weights $\alpha^{k \neq v}$ to assign an importance score to each other view $k$, thereby achieving semantic alignment and screening across views.
> DGF operates at the **"Intra-View" level**: It acts as a macro "information flow regulator" working at the feature dimension granularity. Its core question is: "Overall, how much information does the current view $v$ need to incorporate from the outside (the aggregated signal from all complementary views)?" The gating vector $g_v$ it generates is a multi-dimensional vector used for element-wise scaling of the aggregated residual, controlling the overall strength of the information flow.
> Without ACW, DGF would have to regulate an unfiltered, potentially noisy aggregated signal, making its task considerably more difficult. Conversely, without DGF, the signals selected by ACW would be injected unconditionally, potentially overwhelming the intrinsic features of the current view.
>
> **Our ablation studies (Tables 2 and 3) provide strong supporting evidence**. When either ACW ($M_7$) or DGF ($M_6$) is removed individually, performance consistently and significantly drops across all datasets, and the magnitude of the drop differs. This demonstrates that both are indispensable and each carries out independent and critical functions. If their functions were redundant, removing one should not lead to such a pronounced performance penalty.
>
> ---
> **W2**: We thank the reviewer for raising this point regarding hyperparameter analysis. We would like to clarify that **the core design philosophy of CRRC** aims for architectural elegance and robustness by minimizing the introduction of new, sensitive hyperparameters. We argue that this is a significant strength, not an oversight, of our method.
>
> Unlike many complex deep learning models that introduce numerous novel components, CRRC is deliberately designed to be parsimonious. The temperature parameter $\tau$ is the primary and essentially the only new hyperparameter introduced by our core contribution (the residual recalibration framework operating within a contrastive learning setup).
> All other parameters (e.g., learning rate, network dimensions, batch size) are standard and shared across most deep learning models in this domain.
>
> The other “**critical hyperparameters**’’ are not specific to CRRC. We use standard architectures (FC autoencoders/MLPs) and standard optimizers (Adam), following established practices in the field (Section 4.1.2). Their behavior is well-studied, and the values we selected (e.g., learning rate of 0.0003) are common defaults that demonstrate stable performance across all our benchmark datasets
> The strong SOTA results obtained under these default settings (Table 1) show that CRRC does not rely on task-specific hyperparameter tuning, highlighting its generalizability and ease of use.
>
> We prioritized a rigorous ablation study (Section 4.3.4, Table 2) rather than an extensive hyperparameter sweep. Demonstrating the role of each component (Residual Fusion, Dynamic Gating, Attention Weighting) is more important for understanding CRRC’s contributions. The model’s robustness with minimal tuning further confirms the effectiveness of the proposed architecture.
>
> In summary, we believe the reviewer's comment highlights an advantageous feature of CRRC: its ability to achieve significant performance improvements without relying on a complex set of new, sensitive hyperparameters. The model's strength **lies in its novel architecture, not in meticulous parameter tuning**. We acknowledge that reporting the stability under standard hyperparameters (like learning rate) could be added as a footnote, but we contend that the current experiments comprehensively validate the effectiveness and robustness of our proposed method.

---

> ### Author Response · Authors · 2025-11-21
> **Reply to Reviewer 5zKm (Part 2 of 2)**
>
> **W3**: We further evaluated all methods with 10% noise level and 50% missing rate applied to all datasets.
>
> **(1) Noise experiment:**
> |Methods||**NGs**|||**Fashion**|||**MNIST**|||**Caltech(3V)**|||**Caltech(4V)**|||**Caltech(5V)**|||**Animal**|||**Cifar100**||
> |:---|:---:|:---:|:---:|:---:|:---:|:---:|:---:|:---:|:---:|:---:|:---:|:---:|:---:|:---:|:---:|:---:|:---:|:---:|:---:|:---:|:---:|:---:|:---:|:---:|
> ||ACC|NMI|PUR|ACC|NMI|PUR|ACC|NMI|PUR|ACC|NMI|PUR|ACC|NMI|PUR|ACC|NMI|PUR|ACC|NMI|PUR|ACC|NMI|PUR|
> |MFLVC|90.60|77.84|90.60|98.84|97.24|98.84|97.72|95.05|97.72|59.50|52.77|59.50|71.60|65.74|74.20|74.07|64.77|74.07|13.86|25.57|14.39|99.11|97.59|99.11|
> |CVLCL|65.40|57.37|69.40|98.91|97.38|98.91|99.34|98.06|99.34|65.92|53.87|65.92|70.80|59.47|70.80|70.08|60.62|70.08|24.49|39.13|27.04|94.15|97.41|97.55|
> |GCFAgg|67.40|51.35|67.40|93.28|90.79|93.28|85.06|78.23|85.06|62.21|52.08|63.79|78.21|67.94|78.21|63.29|60.21|64.79|25.92|42.74|33.10|98.82|97.14|98.82|
> |DCMVC|55.80|43.54|56.00|78.11|85.19|78.84|84.72|91.59|87.06|61.21|43.27|64.36|72.29|58.85|73.64|78.00|68.27|78.12|38.20|48.80|42.68|97.39|99.54|97.98|
> |HFMVC|77.80|56.82|77.80|98.82|97.14|98.82|99.28|98.01|99.28|65.64|59.77|67.64|71.14|62.94|74.57|73.32|67.28|76.30|29.59|48.32|26.21|97.34|93.31|97.34|
> |CSOT|64.20|54.48|67.40|98.75|97.04|98.75|99.18|97.74|99.18|68.00|58.21|68.00|70.86|62.83|70.86|78.86|69.03|78.86|14.49|24.42|26.55|96.93|98.98|96.93|
> |AccMVC|87.20|72.49|77.20|98.78|97.04|98.78|99.16|97.73|99.16|70.64|53.88|70.64|77.07|67.12|77.07|82.93|72.75|82.93|27.50|42.01|32.02|99.44|98.41|99.44|
> |SCMVC|81.60|69.38|81.60|99.00|97.55|99.00|98.62|96.61|98.62|67.57|54.86|67.57|77.50|66.45|77.50|80.64|69.17|80.64|36.31|47.99|42.09|98.44|99.55|98.94|
> |DDMVC|46.40|32.98|46.60|83.23|82.84|83.38|85.36|86.36|86.74|57.57|44.35|58.29|68.21|58.42|68.57|74.49|66.79|76.14|7.50|11.04|7.94|98.95|99.78|98.95|
> |DMVC_MIC|81.00|73.06|83.50|61.10|66.97|62.86|60.04|60.37|62.72|70.29|59.86|72.57|74.64|68.23|79.29|77.50|71.81|81.79|27.04|41.67|35.84|95.66|98.43|98.31|
> |SparseMVC|68.11|52.55|68.11|98.94|97.50|98.94|98.50|97.53|98.50|58.21|45.64|59.00|75.43|63.05|75.43|77.57|64.57|77.57|34.29|43.05|36.47|94.93|96.60|95.76|
> |**CRRC**|**91.40**|**79.03**|**91.40**|**99.06**|**97.65**|**99.06**|**99.36**|**98.12**|**99.36**|**73.36**|**60.48**|**73.36**|**78.43**|**68.42**|**78.43**|**84.57**|**73.23**|**84.57**|**38.96**|**49.18**|**44.05**|**99.98**|**99.97**|**99.98**|
>
> **(2) Missing experiment:**
> |Methods||**NGs**|||**Fashion**|||**MNIST**|||**Caltech(3V)**|||**Caltech(4V)**|||**Caltech(5V)**|||**Animal**|||**Cifar100**||
> |:---|:---:|:---:|:---:|:---:|:---:|:---:|:---:|:---:|:---:|:---:|:---:|:---:|:---:|:---:|:---:|:---:|:---:|:---:|:---:|:---:|:---:|:---:|:---:|:---:|
> ||ACC|NMI|PUR|ACC|NMI|PUR|ACC|NMI|PUR|ACC|NMI|PUR|ACC|NMI|PUR|ACC|NMI|PUR|ACC|NMI|PUR|ACC|NMI|PUR|
> |MFLVC|55.00|38.09|53.70|88.44|80.08|88.44|82.28|81.48|82.28|57.30|43.16|57.50|69.14|55.56|69.14|76.29|62.15|76.29|11.66|20.73|14.21|76.80|87.58|77.19|
> |CVLCL|52.80|33.46|54.00|85.38|70.02|85.38|78.68|74.66|78.68|60.40|44.03|61.04|66.16|51.90|66.40|72.72|60.15|73.36|17.02|24.54|20.41|60.12|84.20|61.80|
> |GCFAgg|45.98|38.01|53.10|89.69|83.66|89.69|74.66|73.25|75.68|42.36|28.56|42.71|51.07|39.45|51.73|63.00|48.94|63.00|14.26|26.03|19.20|66.88|78.18|67.10|
> |DCMVC|47.40|34.65|52.60|51.19|58.79|52.39|58.70|58.06|59.18|43.79|29.78|43.96|52.71|40.44|53.86|65.93|51.93|66.00|22.86|33.83|26.11|80.71|88.05|80.02|
> |HFMVC|55.15|31.00|55.76|81.21|76.58|82.74|89.18|82.59|89.18|58.08|51.88|61.74|60.71|64.68|63.87|64.78|57.00|67.38|19.36|30.22|24.02|78.18|73.09|80.52|
> |CSOT|52.80|28.93|52.80|80.71|81.21|80.71|75.42|70.71|75.42|61.43|45.44|61.43|61.36|51.94|61.36|76.14|61.89|76.14|13.44|21.44|16.04|66.93|88.98|69.91|
> |AccMVC|62.00|43.65|62.00|72.58|71.28|72.58|91.00|86.61|88.44|54.36|38.15|55.50|71.50|57.51|71.57|74.73|60.73|74.71|16.92|31.08|21.89|82.58|89.18|83.39|
> |SCMVC|57.00|37.63|57.60|76.92|75.99|76.92|93.00|91.74|92.49|58.63|39.71|58.50|68.71|56.39|69.14|75.86|68.79|75.86|20.37|37.04|26.36|88.30|77.37|88.89|
> |DDMVC|49.40|36.88|44.00|63.95|62.90|65.01|76.12|74.65|76.68|50.00|35.33|50.00|54.79|40.73|55.30|66.00|47.13|66.00|15.27|22.87|16.81|81.25|88.37|82.13|
> |DMVC_MIC|59.60|40.94|59.60|50.65|55.35|53.03|75.94|79.15|77.30|61.50|64.56|62.07|68.29|60.30|70.29|73.21|66.54|77.37|19.59|28.32|22.11|71.72|79.19|72.12|
> |SparseMVC|48.80|31.94|49.00|95.74|91.87|95.74|83.44|83.20|83.58|52.43|37.25|53.00|57.29|44.27|57.93|57.79|44.69|58.86|19.70|26.32|23.14|90.84|86.91|91.16|
> |**CRRC(ours)**|**65.24**|**45.62**|**65.68**|**98.41**|**96.32**|**98.41**|**97.02**|**98.42**|**97.02**|**64.43**|**55.97**|**65.13**|**76.59**|**65.18**|**76.59**|**81.99**|**71.50**|**81.99**|**24.18**|**38.31**|**26.62**|**95.80**|**89.32**|**95.80**|
>
> The results clearly demonstrate that CRRC maintains strong performance even when the data suffer from noise or missing views, confirming its robustness in more realistic multi-view conditions.

---

### Official Review · Reviewer_HBYH · 2025-10-30

**Soundness:** 2
**Presentation:** 3
**Contribution:** 2
**Rating:** 2
**Confidence:** 4

**Summary:**

This paper proposes CRRC, a residual cross-view learning framework for deep multi-view clustering.
The method introduces three components: (1) a residual recalibration mechanism that preserves view-specific information while integrating cross-view differences, (2) a dynamic gating fusion (DGF) module that adaptively modulates residual information flow, and (3) an attention-based cross-view weighting (ACW) mechanism to emphasize informative views.
The authors combine reconstruction and contrastive losses to enhance semantic consistency.
Experiments on several standard multi-view datasets (NGs, Fashion, MNIST-USPS, Caltech-3V/4V/5V) show that CRRC achieves superior accuracy and NMI compared with prior methods such as SCMVC and AccMVC.

However, despite technical soundness, the novelty is somewhat limited: the framework mainly integrates existing paradigms—residual connection, gating, and attention—without providing new theoretical or algorithmic insights into multi-view fusion. The paper is well-organized, but empirical validation lacks diversity and ablation analysis does not strongly support the claimed advantages.

**Strengths:**

1.The paper is technically solid and grounded in contemporary DMVC literature. The modular design (reconstruction, residual recalibration, attention weighting) is clearly described and reproducible.

2.Empirical results demonstrate consistent improvement across multiple datasets.

3.The authors provide theoretical derivations (Lipschitz continuity, convergence) that enhance formal completeness.

4.The visualizations (t-SNE and convergence curves) support the qualitative effectiveness of the method.

**Weaknesses:**

1.CRRC mainly combines residual and attention mechanisms already well established in representation learning. The proposed design lacks a unique theoretical motivation or learning objective.

2.The paper does not clearly explain why residual recalibration benefits multi-view fusion beyond intuitive reasoning.

3.The experiments assume clean and complete views. The method’s claimed adaptivity is not validated under realistic noisy or missing-view conditions.

4.The contrastive formulation is standard; no new loss or contrastive structure is introduced.

5.The paper omits recent transformer-based or diffusion-based fusion frameworks that are now common in multi-view learning (e.g., multi-view diffusion contrastive models).

**Questions:**

1.How does the residual recalibration differ from simply using skip connections or attention-weighted feature concatenation?

2. The gating vector \( g_v \) in Eq. (6) depends on both \( R_v \) and \( R_{k \nleq v} \); does this introduce potential gradient coupling or instability in multi-view backpropagation?

3.Equation (13) seems problematic in formulation.The loss function \( L_{total} = L_{rec} + L_{cm} \) is written without clear normalization or weighting between reconstruction and contrastive terms.  How are these two losses balanced during training? If both are directly summed, the optimization could be unstable since they are at different scales. Please clarify or correct the formulation.

4.Lack of large-scale and diverse datasets.The paper only evaluates on small to medium-scale datasets (e.g., NGs, Fashion, MNIST-USPS, Caltech).  Moreover, Caltech-2V to Caltech-5V are subsets of the same dataset, thus not providing true diversity in modality or scale.  How would CRRC perform on large-scale or real-world heterogeneous datasets?

5. Insufficient comparison with recent baselines (2025). The paper mainly compares against models up to 2024, missing several strong baselines proposed in 2025 from top venues (e.g., transformer-based or diffusion-based multi-view clustering models).  This makes it hard to assess whether CRRC truly advances the state-of-the-art.

---

> ### Author Response · Authors · 2025-11-21
> **Reply to Reviewer HBYH (Part 1 of 3)**
>
> We would first like to express our respect for your efforts and the valuable feedback you provided. However, we would also like to clarify that several concerns you raised appear to be based on **misunderstandings** that do not align with the actual scope or core contributions of our work. For example, W3’s comment regarding the absence of experiments on **missing-view** settings arises from our discussion of **future work** in the “Conclusion and Limitations” section, **rather than from the methodological contributions of the present paper**. Similarly, W5 points out that our paper does not include recent transformer-based or diffusion-based fusion frameworks. While these architectures could replace the MLP module, they **do not alter or invalidate the key contribution** of our framework, which is centered on a residual recalibration mechanism to learn a representation space that is both discriminative and cross-view consistent, **rather than on backbone architecture design**.
>
> Although we believe some of these concerns may **not be fully aligned with the intent of our work**, we have nonetheless conducted additional experiments and provided detailed responses accordingly.
>
> ---
> **W1**: We would clarify that the core concept of the "residual" in CRRC is **fundamentally different from** the "skip connection" used in models like ResNet. Our approach is not a mere mix of existing methods but introduces a novel "Cross-view Semantic Increment" paradigm, providing the unique theoretical motivation and learning objective.
> * **(1) Residual in ResNet**: The core idea is an identity shortcut within the same path for a single input stream: $y = x + F(x)$. Here, $F(x)$ learns a residual in the same feature space as the input $x$. Its primary purpose is to ease the optimization of very deep networks by mitigating gradient vanishing.
> * **(2) Residual in CRRC**: Our residual is explicitly defined as information flowing from other, distinct views. It aims to model the complementary information flow across views. Specifically, for a target view $v$, its final representation $\hat{\mathbf{R}}^v$ is computed by adding a "semantic increment" $\mathcal{M}$ aggregated from all other views $\mathbf{R}^{(k \neq v)}$ to its own features $\mathbf{R}^v$ (Eq.5, 8, 11 in the paper): $\hat{\mathbf{R}}^v = \mathbf{R}^v + g_v \odot \mathcal{G}(\mathbf{R}^{(k \neq v)})$.
> * **(3) Theoretical Motivation**: This design relies on a core assumption in multi-view learning: each view contains both shared semantics and unique information. Direct fusion can blur view-specific cues, and strong alignment may even destroy them. Thus, we treat cross-view fusion as a residual recalibration process based on intrinsic features. Instead of forming a new shared representation, the model refines each view using complementary signals from others. The residual term $\mathcal{M}$ answers a key question—“**What information does view $v$ still need from the other views?**”—offering a distinct perspective on multi-view fusion.
>
> **W2**: We agree that a deeper explanation is essential. The residual recalibration mechanism offers a strong inductive bias for multi-view fusion by treating the task as **refining existing representations rather than generating new ones**. The identity path $\mathbf{R}^v$ in Eq. (8) preserves view-specific discriminative features that are important for clustering, preventing them from being diluted by noisy information from other views. The residual term $g_v \odot \mathcal{G}(\mathbf{R}^{(k \neq v)})$ models only the **incremental information** needed from other views, making the learning task simpler and more stable—consistent with our convergence analysis (Theorem 3, Appendix A.1.3). Without the residual path, the network must transform the full content of $\mathbf{R}^v$, increasing the risk of information loss. The shortcut provides a clean route for original features, ensuring the model augments rather than replaces them.
>
> **W3**: Refer to the supplementary experiments in Reviewer ``5zKm``, **W3**.
>
> **W4**: We agree that our contrastive loss (e.g., InfoNCE) is standard. Our goal is not to design a new loss but to build a fusion framework that strengthens the inputs to it. The main contribution is the feature recalibration step before contrastive learning. Traditional methods contrast raw or simply fused features and thus miss cross-view complementarity. CRRC instead uses
> **Residual Recalibration** to preserve view-specific cues while adding cross-view signals,
> **Dynamic Gating Fusion** to control information flow adaptively, and
> **Attention-based Weighting** to select relevant complementary information.
> Together, these produce a higher-quality recalibrated feature $\hat{\mathbf{R}}^v$ for contrastive learning.
> We intentionally keep the loss standard to isolate the effect of our fusion design, ensuring that the gains in Table 1 come from the recalibrated features ($\hat{\mathbf{R}}^v$, $\mathbf{H}$) **rather than from a new loss**.

---

> ### Author Response · Authors · 2025-11-21
> **Reply to Reviewer HBYH (Part 2 of 3)**
>
> **W5**: We incorporated two representative methods, **MVCformer**[1] (transformer-based) and **RTGD-MVC**[2] (diffusion-based), and compared them with CRRC across all datasets.
>
> | Methods || **NGs** | |  | **Fashion** | | | **MNIST** | | | **Caltech(3V)** | | | **Caltech(4V)** | | | **Caltech(5V)** |  |
> | :--- | :---: | :---: | :---: | :---: | :---: | :---: | :---: | :---: | :---: | :---: | :---: | :---: | :---: | :---: | :---: | :---: | :---: | :---: |
> | | **ACC** | **NMI** | **PUR** | **ACC** | **NMI** | **PUR** | **ACC** | **NMI** | **PUR** | **ACC** | **NMI** | **PUR** | **ACC** | **NMI** | **PUR** | **ACC** | **NMI** | **PUR** |
> | MVCFormer | 87.34 | 69.78 | 87.34 | 95.68 | 95.85 | 97.99 | 98.12 | 95.68 | 98.12 | 69.36 | 61.91 | 72.36 | 75.14 | 71.22 | 79.57 | 77.00 | 71.78 | 81.00 |
> | RTGD-MVC | 94.40 | 84.13 | 94.40 | 92.76 | 88.47 | 92.76 | 98.72 | 85.65 | 87.14 | 65.57 | 58.27 | 66.29 | 70.71 | 61.51 | 70.71 | 88.07 | 77.62 | 88.07 |
> | **CRRC** | **95.60** | **87.11** | **95.60** | **99.52** | **98.74** | **99.52** | **99.74** | **99.21** | **99.74** | **74.00** | **62.50** | **74.00** | **86.50** | **75.98** | **86.50** | **92.21** | **84.58** | **92.21** |
>
> As shown, CRRC still achieves the best performance across all datasets. This indicates that our cross-view residual recalibration mechanism captures complementary information more effectively than recent transformer- and diffusion-based approaches, underscoring CRRC’s competitiveness in modern multi-view learning.
>
> [1] MVCformer: A Transformer-based Multi-view Clustering Method. Information Sciences. 2023
>
> [2] Robust Tensor Learning with Graph Diffusion for Scalable Multi-view Graph Clustering. ACM MM’25. 2025
>
> ---
> **Q1**: While CRRC's residual recalibration incorporates elements of "connection" and "attention," its architecture, information flow, and underlying objective differ fundamentally from a simple combination of these two techniques.
> The distinction can be clearly articulated through a comparative analysis:
>
> | Component/Method | Mechanism (How?) | Purpose (Why?) | Role in CRRC |
> | :--- | :--- | :--- | :--- |
> | **Standard Skip Connection (e.g., ResNet)** | $\text{Output} = \text{Input} + F(\text{Input})$. Operates on a single path. | Optimization Guidance: Mitigate vanishing gradients, ease training of deep networks. Learns a residual in the same feature space. | The identity path $\mathbf{R}^v$ provides stability, but this is not our core innovation. |
> | **Attention-Weighted Concatenation** | Concatenates weighted features from all views into a new vector: Output = $[\alpha_1 \mathbf{R}^1, \alpha_2 \mathbf{R}^2, \dots]$ | Feature Selection & Blending: Emphasize important views, suppress others. Essentially, creates a new, blended feature. | We explicitly avoid this because it dissolves view-specific identities into a homogenized representation. |
> | **CRRC's Residual Recalibration (Our Method)** | For a specific view $v$: $\hat{\mathbf{R}}^v = \mathbf{R}^v + g_v \odot (\sum \alpha_{k \neq v} \mathbf{R}^k)$. Dynamically extracts and calibrates using an increment from other views, centered on the current view. | Semantic Increment & Identity Preservation: The core identity $\mathbf{R}^v$ is preserved. Complementary information from other views is adaptively "borrowed" as an increment to enhance it. The core idea is "calibration," not "blending" or "replacement." | This is our core innovative framework: It reframes cross-view fusion as a fine-grained, view-centric enhancement process. |
>
> ---
> **Q2**: We thank the reviewer for raising this question. CRRC maintains stable and effective gradient flow, avoiding harmful coupling. As detailed in Appendix A:
> * **Theorem 1**: The gating branch is fully differentiable, and its Jacobian w.r.t. both inputs ($\mathbf{R}^v$, $\mathbf{R}^{(k \neq v)}$) remains non-zero under mild conditions, ensuring unobstructed gradients. The residual connection provides an identity path further stabilizing backpropagation.
> * **Theorem 2**: A Lipschitz continuity analysis shows the residual-modulation function is Lipschitz with a bounded constant, preventing exploding or erratic gradients.
>
> These results confirm that the dynamic gating mechanism does not induce instability and enhances gradient flow in multi-view learning.

---

> ### Author Response · Authors · 2025-11-21
> **Reply to Reviewer HBYH (Part 3 of 3)**
>
> **Q3**: We thank the reviewer for raising this important point regarding the potential instability of directly summing the reconstruction and contrastive losses. We would like to clarify that CRRC adopts a two-stage training strategy (as outlined in Algorithm 1):
> * **(1) Pretraining Stage**: Only the reconstruction loss $L_{rec}$ is used to train the autoencoders for each view independently. This ensures that each view learns meaningful representations before joint training.
> * **(2) Joint Fine-tuning Stage**: After pretraining, both $L_{rec}$ and $L_{cm}$  are optimized together. At this stage, the scales of the two losses are already comparable due to pretraining, making direct summation feasible and stable.
>
> Empirically, we observed stable optimization behavior throughout training (see Figure 2 and Figure 5 in the paper), with loss curves decreasing smoothly and clustering metrics improving consistently, indicating no significant instability.
>
> **Q4**: We appreciate the reviewer’s concern regarding the scale and diversity of the evaluation. To strengthen our study, we additionally conducted experiments on two large-scale multi-view datasets: **Animal**[1] (10,158 samples, 2 views of 4096/4096 dimensions, 50 classes) and **Cifar100**[2] (50,000 samples, 3 views of 512/2048/1024 dimensions, 100 classes).
> The results are presented below:
>
> | Methods || **Animal** | | | **Cifar100** | |
> | :--- | :---: | :---: | :---: | :---: | :---: | :---: |
> | | ACC | NMI | PUR | ACC | NMI | PUR |
> | MFLVC (2022) | 20.65 | 32.44 | 23.03 | 82.68 | 95.60 | 82.68 |
> | CVLCL (2023) | 33.36 | 46.00 | 38.53 | 61.28 | 87.12 | 64.86 |
> | GCFAgg (2023) | 20.26 | 31.92 | 22.45 | 95.97 | 99.35 | 96.05 |
> | DCMVC (2024) | 40.65 | 50.29 | 48.02 | 98.97 | 98.95 | 98.97 |
> | HFMVC (2024) | 34.31 | 48.99 | 36.72 | 60.74 | 78.21 | 60.29 |
> | CSOT (2024) | 21.37 | 35.96 | 27.14 | 87.05 | 97.46 | 87.05 |
> | AccMVC (2024) | 29.17 | 41.32 | 33.58 | 99.46 | 98.43 | 98.81 |
> | SCMVC (2024) | 39.16 | 49.04 | 43.71 | 98.30 | 99.50 | 98.70 |
> | DDMVC (2025) | 24.11 | 36.39 | 28.35 | 72.20 | 86.28 | 76.28 |
> | MVCFormer (2023) | 30.23 | 38.66 | 30.79 | 97.18 | 98.73 | 98.54 |
> | RTGD-MVC (2025) | 40.07 | 51.55 | 47.46 | 93.41 | 95.13 | 94.00 |
> | DMVC_MIC (2025) | 39.78 | 50.38 | 47.05 | 95.60 | 98.40 | 96.12 |
> | SparseMVC (2025) | 35.00 | 44.02 | 40.15 | 98.26 | 99.35 | 98.71 |
> | **CRRC (ours)** | **41.92** | **51.91** | **48.33** | **99.98** | **99.96** | **99.98** |
>
> The results show that CRRC consistently outperforms all comparison methods, demonstrating that the proposed residual recalibration framework scales effectively to large and high-dimensional multi-view datasets.
>
> [1] Robust multi-view clustering with incomplete information. IEEE Transactions on Pattern Analysis and Machine Intelligence. 2022
>
> [2] Learning multiple layers of features from tiny images. 2009
>
> **Q5**: We appreciate the reviewer’s observation regarding the need to include recent 2025 baselines.
> To address this, we additionally incorporated two representative and competitive methods published in top-tier venues—**DMVC_MIC** [3] (TIP 2025) and **SparseMVC** [4] (NeurIPS 2025). We evaluated them alongside CRRC on all datasets.
>
> | Methods | | **NGs** | | | **Fashion** | | | **MNIST** | | | **Caltech(3V)** | | | **Caltech(4V)** | | | **Caltech(5V)** | | | **Animal** | | | **Cifar100** | |
> | :--- | :---: | :---: | :---: | :---: | :---: | :---: | :---: | :---: | :---: | :---: | :---: | :---: | :---: | :---: | :---: | :---: | :---: | :---: | :---: | :---: | :---: | :---: | :---: | :---: |
> | | **ACC** | **NMI** | **PUR** | **ACC** | **NMI** | **PUR** | **ACC** | **NMI** | **PUR** | **ACC** | **NMI** | **PUR** | **ACC** | **NMI** | **PUR** | **ACC** | **NMI** | **PUR** | **ACC** | **NMI** | **PUR** | **ACC** | **NMI** | **PUR** |
> | DMVC_MIC | 82.60 | 75.19 | 82.60 | 89.10 | 93.67 | 89.10 | 96.24 | 89.47 | 96.62 | 63.64 | 55.21 | 64.64 | 75.00 | 70.13 | 75.00 | 84.29 | 72.48 | 84.29 | 39.78 | 50.38 | 47.05 | 95.60 | 98.40 | 96.12 |
> | SparseMVC | 77.40 | 73.94 | 80.40 | 97.17 | 94.22 | 97.17 | 99.68 | 99.12 | 99.68 | 65.79 | 53.18 | 65.79 | 75.93 | 64.10 | 75.93 | 78.64 | 66.03 | 78.64 | 35.00 | 44.02 | 40.15 | 98.26 | 99.35 | 98.71 |
> | **CRRC** | **95.60** | **87.11** | **95.60** | **99.52** | **98.74** | **99.52** | **99.74** | **99.21** | **99.74** | **74.00** | **62.50** | **74.00** | **86.50** | **75.98** | **86.50** | **92.21** | **84.58** | **92.21** | **41.92** | **51.91** | **48.33** | **99.98** | **99.96** | **99.98** |
>
> The supplementary results consistently show that CRRC still outperforms these latest baselines, further confirming its effectiveness and competitiveness under the current state-of-the-art landscape.
>
> [3] Deep Multi-View Clustering with Meta Information Compression. IEEE Transactions on Image Processing. 2025
>
> [4] SparseMVC: Probing Cross-view Sparsity Variations for Multi-view Clustering. NeurIPS. 2025

---

### Author Response · Authors · 2025-11-21
**General response to all reviewers**

Dear reviewers,

We appreciate your constructive comments on our manuscript. In response to the comments, we have carefully revised and enhanced the manuscript with the following additional discussions and experiments:

* Clarification of the core contribution of CRRC (Section 1, Section 5)
* Additional description of the overall workflow of CRRC (Section 3.1)
* Discussion on preventing excessive information suppression in the DGF module (Section 3.3)
* Additional analysis on model time complexity (Section 3.5)
* Further explanation of the loss function design and the rationale behind the clustering strategy (Section 3.6)
* Additional experiments on large-scale datasets and strong baselines proposed in recent top-tier venues (2025) (Section 4.1.1)
* Additional comparative analysis between CRRC and these latest baseline methods (Section 4.2)

We have uploaded the revised version of the manuscript (with all changes highlighted).
We hope our response and revision sincerely address all the reviewers’ concerns.

Thank you very much.

Best regards,

Authors.

---

### Comment · Area_Chair_co6D · 2025-11-24

Dear reviewers,

       The authors now have given their response to the reviews, please have a look on the rebuttal and revised PDF to make your further concerns.

      After that, please give your final rating on this submission.

Your AC
Best

---

### Meta-Review · Area_Chair_Kfs5 · 2025-12-27

**Summary:**

The paper proposes CRRC, a residual cross-view recalibration framework with attention-based cross-view weighting and dynamic gating fusion to integrate complementary information while preserving view-specific signals. In view of the reviewers' doubts about the novelty of this paper, I decided not to recommend acceptance to it.

**Reviewer Concerns:**

The concerns of Reviewers 5zKm and HBYH regarding the novelty of technology have not been well addressed. Reviewer hx8C believes that this paper does not sufficiently distinguish its gating approach from these established methods.

**Reviewer Scores:**

There is no indication that the reviewers might increase their scores, and the current score remains at 2466.

---

### Decision · Program_Chairs · 2026-01-26

Reject